# Pushover-Based Seismic Capacity Evaluation of Uto City Hall Damaged by the 2016 Kumamoto Earthquake

**Kenji Fujii**

Department of Architecture, Faculty of Creative Engineering, Chiba Institute of Technology, Chiba 275-0016, Japan; kenji.fujii@it-chiba.ac.jp

**Abstract:** The seismic capacity of the main Uto City Hall building, which was severely damaged by the 2016 Kumamoto Earthquake, was evaluated by the proposed pushover-based procedure. In this procedure, the seismic capacity index of the building is defined as the maximum scaling factor of the seismic input, for which the local responses do not exceed their limit values. From the pushover analysis result, the displacement limit of the equivalent single-degree-of-freedom model was determined. Then, the seismic capacity index was evaluated using an equivalent linearization technique. The evaluated index was re-evaluated by considering the bidirectional excitation. The pushover analysis result revealed that the torsional response is significant in the nonlinear behavior of this building. The evaluated seismic capacity implied that some structural damages, including the yielding of the beam-column joint, may have occurred during the first earthquake on 14 April 2016.

**Keywords:** existing building; structural irregularity; pushover analysis; bidirectional excitation; seismic capacity evaluation; torsion

## 1. Introduction

### 1.1. Background

Buildings with highly irregular plans and/or elevations tend to be vulnerable to earthquakes. In the 2016 Kumamoto Earthquake, the main building of Uto City Hall was severely damaged [1]. This building is a five-story reinforced concrete (RC) irregular building, which was constructed in 1965. Given that the buildings around the city hall did not incur severe damages, it can be said that the main building had low seismic performance owing to structural irregularities and old construction methods based on an inadequate seismic design code. The author had carried out the preliminary seismic evaluation of this building based on the structural drawing [2]. The preliminary evaluation results had revealed that the seismic capacity of this building was insufficient to survive severe earthquakes. However, the evaluated results could not explain the damage of the building observed in upper stories. One of the reasons is that the effect of torsion is not considered in the simplified method applied for the preliminary evaluation.

Nowadays, the available simplified nonlinear analysis procedures combine the nonlinear static (pushover) analysis of a multi-degree-of-freedom (MDOF) model with the response spectrum analysis of an equivalent single-degree-of-freedom (SDOF) model [3–5]. These procedures have been widely implemented in seismic design codes and seismic design schemes [6–9], and work well for buildings that oscillate predominantly with a single mode. Although these simplified procedures are less accurate than the more sophisticated nonlinear time-history analysis, they provide us with basic information about the building under investigation. Thereby, they assist designers and analysts in understanding

the nonlinear seismic behavior of buildings. In recent years, the author and other researchers have worked to extend these simplified procedures and improve the seismic performance estimates of buildings with plan and/or elevation irregularities [10–29].

These extended simplified procedures have been verified by comparison to nonlinear dynamic analysis results. However, it is challenging to verify them using existing irregular buildings that were damaged during an actual seismic event. Therefore, verifying these simplified procedures by considering the main Uto City Hall building is an important benchmark of their ability.

*1.2. Brief Review of Related Studies*

The author has determined that there are four possible approaches toward predicting the peak response of an asymmetric building with consideration given to the torsional effects. The first one is *the extended N2 method* [10–14]; the second one is a modal pushover analysis [15–22]; the third one consists of combining two pushover analyses and the envelope of the results, as proposed by Bosco et al. [23,24]; the fourth one consists of combining the analyses of two independently equivalent SDOF models (representing the first and second modes) with the envelope of four pushover analyses (including the effect of bidirectional excitation) [25–29].

The first approach, namely, *the extended N2 method*, is an extended version of the simplified procedure proposed by Fajfar and Fischinger [4]. This approach involves the estimation of the peak response for each frame in the pushover analysis results multiplied by a correction factor, which is defined using the linear dynamic analysis and pushover analysis results [10–13].

The second approach, namely, modal pushover analysis (MPA), was proposed by Chopra and Goel for regular buildings and considers the effect of higher modes. Subsequently, this approach was extended to asymmetric buildings [15]. Then, it was extended further by Reyes and Chopra [16,17], who considered the effect of bidirectional excitation. Manoukas et al. [18,19] proposed the concept of an equivalent SDOF model, which differed from that proposed by Chopra. Manoukas et al. considered the effect of bidirectional excitation in the formulation of the equivalent SDOF model. Belejo and Bento [20] applied the improved pushover analysis (IMPA) (a modified version of MPA that considers mode shape changes in the inelastic regime) to buildings with three and nine stories. Another interesting variant of MPA based on energy-based pushover analysis has been proposed by Soleimani et al. [21].

As proposed by Bosco et al. [23], the third approach estimates the peak response of the frames on the stiff side and the peak response of the frames on the flexible side by enveloping the results of two pushover analyses. Bosco et al. [24] investigated the applicability of this procedure to multistory building models with the same geometry on each floor. In this procedure, corrective eccentricity is a key parameter in the pushover analyses.

The fourth approach has been proposed by the author [25–29]. In this paper, this simplified procedure is called mode-adaptive bidirectional pushover analysis (MABPA), and it involves the evaluation of the first and second mode peak responses independently from the equivalent SDOF models formulated with consideration given to the principal direction of the first modal response. The prediction of the peak response at each frame is based on a set of pushover analyses that consider the combination of the two modal responses. Although the latest version of MABPA [26] can successfully predict the peak response of asymmetric buildings, it requires two critical assumptions. The first assumption is that the building oscillates predominantly in a single mode and in each set of orthogonal directions. The second assumption is that the principal directions of the first and second modal responses are almost orthogonal. The limitation of the current version of MABPA has been discussed in a previous report [29].

These studies mentioned above were mainly focused on the prediction of the peak response (mainly displacement) under given ground motions (mostly by the response spectrum). Some of these procedures may be used for the prediction of the seismic intensity, or scaling factor, corresponding to a certain (given) displacement limit. Dolšek and Fajfar [14] proposed a simplified probabilistic seismic performance assessment for plan-asymmetric buildings. In their study, the ground motion intensity at

the failure point was evaluated using *the extended N2 method*, so-called the incremental N2 method. Also, as an application of MPA, a scaling procedure for the nonlinear dynamic analysis of asymmetric buildings with consideration given to the bidirectional excitation was proposed by Reyes et al. [22]. In their procedure, the ground acceleration was scaled such that the peak roof displacement was equal to the target displacement. If the given seismic intensity is too large for the building considered, its peak response would be too far from its limit value: sometimes its peak response may not be obtained because the analysis would be unstable. In such case, the evaluation of the scaling factor of the given seismic input, wherein the local responses do not exceed their limit value, would be helpful for the understanding of the seismic behavior of the building considered.

### 1.3. Objectives

In this study, the seismic capacity of the main Uto City Hall building was evaluated according to MABPA, which has been previously proposed by the author. The seismic capacity index, which is defined in this paper as the seismic scaling factor of the given seismic input, is evaluated considering bidirectional excitation. The verification of MABPA is made by comparing the observed damages of the main Uto City Hall building and the analysis results, especially concerning the effect of torsion to the structural damage.

The rest of the paper is organized as follows. In Section 2, the seismic capacity index discussed in this paper is defined. Subsequently, a description of the seismic capacity index is provided with consideration given to the unidirectional and bidirectional ground motion. In Section 3, basic information is provided with regard to the main Uto City Hall building, its model, and ground motions. In Section 4, the evaluation procedure of the seismic capacity index is verified by using nonlinear dynamic analyses. Finally, in Section 5, the seismic capacity of this building is evaluated, considering the variation of modeling about strength degradation behavior of walls and basement spring. Then the evaluated seismic capacity curves are compared to the response spectrum of the first and second earthquakes, occurred on the 14 and 16 April, respectively, and discuss the performance of this building against two earthquakes.

## 2. Seismic Capacity Evaluation Procedure

### 2.1. Definition of Seismic Capacity Index

A set of orthogonal axes ($\xi$-$\zeta$) in the *X-Y* plane is considered, where the $\xi$- and $\zeta$-axis are the major and minor axes, respectively, of the horizontal ground motion components. Another set of orthogonal axes, namely *U-V* in the *X-Y* plane, is considered, with the *U*- axis being the principal axis of the first modal response [25]. It is assumed that the spectra of the two horizontal components are identical, as expressed by Equation (1).

$$_pS_{A\xi}(T,h) = {_pS_{A\zeta}}(T,h) = {_pS_{AU}}(T,h) = {_pS_{AV}}(T,h) = {_pS_A}(T,h).\tag{1}$$

In Equation (1) $_pS_A(T, h)$ is the pseudo acceleration spectrum, and *T* and *h* are the natural period and damping ratio, respectively, of the elastic SDOF oscillators. Note that this assumption, namely, the identical-component assumption, is the same as discussed by López and Torres [30] with regard to the elastic spectrum analysis. In addition, López et al. [31] reported that the ratio of the spectra of the horizontal minor and major components varies between 0.63 and 0.81. Therefore, the identical-component assumption is expected to provide conservative prediction results.

The seismic capacity index ($C_I$) is defined in Figure 1. The building model is assumed to be converted to the equivalent SDOF model representing the first mode response. The equivalent acceleration $A_{1U}{}^*$-$D_{1U}{}^*$ relationship, which is shown in this figure, will be termed hereafter as the capacity curve. The displacement limit ($D_{1U}{}^*_{limit}$) is defined as the equivalent displacement, wherein one of the local responses, such as the story or column drift and the plastic hinge rotation angle of the beam, reaches the limit value. The pseudo-acceleration spectrum-displacement spectrum ($_pS_A$-$S_D$)

relationship of the design (or code-specified) ground motion is considered, and is also shown in this figure. The factored $_pS_A$-$S_D$ relationship considering the equivalent damping $h_{eq}$ will be termed hereafter as the demand curve. The intersection point of the capacity and the demand curve represents the predicted peak response. The seismic capacity index is defined as the maximum scaling factor, wherein the local responses do not exceed their limit value. In this figure, the capacity index $C_I$ is equal to the scaling factor producing the peak equivalent displacement equal to $D_{1U}{}^*_{limit}$.

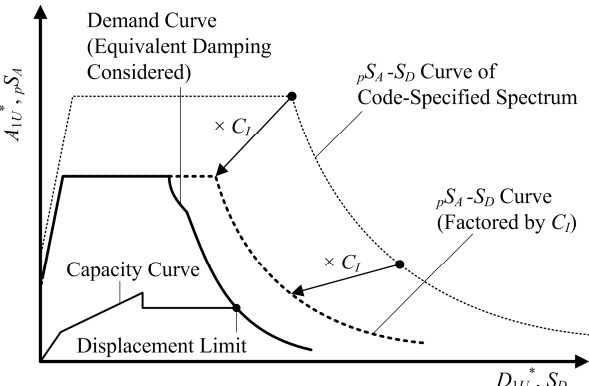

**Figure 1.** Definition of capacity index $C_I$.

In this paper, the seismic capacity index that considers the unidirectional ground motion acting in the principal direction of the first modal response is denoted as $C_{I, uni}$, whereas that considering the bidirectional ground motion acting simultaneously is denoted as $C_{I, bi}$. As described below, $C_{I, uni}$ is evaluated only on the basis of the first modal response, whereas $C_{I, bi}$ is evaluated by using MABPA and by considering the first and second modal responses.

The basic idea of the seismic capacity index $C_I$ is identical: to the ratio (%NBS) defined in the New Zealand (NZ) seismic assessment guideline for existing buildings [32–34]; to the ratio of the ultimate capacity of a building as a whole or of an individual member; to the ultimate limit state shaking demand for a similar new building on the same site. The index $C_{I, uni}$ is identical to the ratio (%NBS) obtained by the nonlinear static pushover analysis by considering the single mode, or obtained by the simple lateral mechanism analysis (SLaMA), the simplified nonlinear static analysis [33,34]. Veccho et al. show an application of the SLaMA method to an existing RC building which has severely damaged during the 2011 Christchurch earthquake [35]. The advantage of the proposed procedure is in considering the bidirectional excitation, which may be more significant for a building with an irregular structural plan.

Note that the presented procedure can be applicable to a building which oscillates predominantly in the first mode under *U*-directional (unidirectional) excitation, as discussed in the previous study [29]. This is because in this procedure the behavior of the building can be represented by the equivalent SDOF model of the first mode, even though the effect of the second mode would be included in the evaluation of $C_{I, bi}$. Therefore, the effective first modal mass ratio of the building should be checked for the applicability of this procedure.

*2.2. Evaluation of the Seismic Capacity Index Considering the Unidirectional Ground Motion*

2.2.1. Step 1: Pushover Analysis of Building Model (First Mode)

The nonlinear equivalent acceleration-equivalent displacement ($A_{1U}{}^*$-$D_{1U}{}^*$) relationship of the equivalent SDOF model was determined based on the pushover analysis of an *N*-story irregular building model considering the change of the first mode's shape at each nonlinear stage.

In this study, the DB-MAP analysis was adopted [26]. The fundamental assumptions of the current version are as follows:

1. The envelope curve of the force-deformation relationship for each nonlinear spring of all members is symmetric in the positive and negative loading direction.
2. The equivalent stiffness of each nonlinear spring can be defined by its secant stiffness at the peak deformation that was previously derived in the calculation.
3. The first mode shape at each loading step ($_n\Gamma_{1U\mathbf{n}}\boldsymbol{\varphi}_1$) can be determined according to the equivalent stiffness.
4. The displacement vector at each loading step ($_\mathbf{n}\mathbf{d}$) imposed on the model is similar to the first mode shape obtained in assumptions 2 and 3.
5. In the case where unloading occurs, the equivalent stiffness obtained in assumption 2 is used as the unloading stiffness for all nonlinear springs.

Note that assumption 5 was adopted from previous studies [26–29] to apply this DB-MAP to a building model with non-ductile members.

Let $_\mathbf{n}\mathbf{f_R}$ be the restoring force vector of the building model at each loading step $n$ obtained from the pushover analysis. The equivalent displacement and acceleration at step $n$ (namely, $_nD_{1U}{}^*$ and $_nA_{1U}{}^*$, respectively) are determined from Equations (2) and (3), respectively, assuming that $_\mathbf{n}\mathbf{d}$ is proportional to the first mode vector $_n\Gamma_{1U\mathbf{n}}\boldsymbol{\varphi}_1$ at each loading step:

$$_nD_{1U}{}^* = \frac{_n\Gamma_{1U\mathbf{n}}\boldsymbol{\varphi}_1{}^\mathbf{T}\mathbf{M_n d}}{_nM_{1U}{}^*} = \frac{\sum_j\left(m_{j\ n}x_j^2 + m_{j\ n}y_j^2 + I_{j\ n}\theta_j^2\right)}{\sqrt{\left(\sum_j m_{j\ n}x_j\right)^2 + \left(\sum_j m_{j\ n}y_j\right)^2}}, \tag{2}$$

$$_nA_{1U}{}^* = \frac{_n\Gamma_{1U\mathbf{n}}\boldsymbol{\varphi}_1{}^\mathbf{T}{}_\mathbf{n}\mathbf{f_R}}{_nM_{1U}{}^*} = \frac{\sum_j\left(_nf_{RXj\ n}x_j + {}_nf_{RYj\ n}y_j + {}_nf_{MZj\ n}\theta_j\right)}{\sqrt{\left(\sum_j m_{j\ n}x_j\right)^2 + \left(\sum_j m_{j\ n}y_j\right)^2}}, \tag{3}$$

$$_nM_{1U}{}^* = {}_n\Gamma_{1U}{}^2{}_\mathbf{n}\boldsymbol{\varphi}_1{}^\mathbf{T}\mathbf{M_n}\boldsymbol{\varphi}_1 = \frac{\left(\sum_j m_{j\ n}x_j\right)^2 + \left(\sum_j m_{j\ n}y_j\right)^2}{\sum_j\left(m_{j\ n}x_j^2 + m_{j\ n}y_j^2 + I_{j\ n}\theta_j^2\right)}, \tag{4}$$

$$_n\Gamma_{1U} = \frac{_\mathbf{n}\boldsymbol{\varphi}_1{}^\mathbf{T}\mathbf{M_n}\boldsymbol{\alpha_U}}{_\mathbf{n}\boldsymbol{\varphi}_1{}^\mathbf{T}\mathbf{M_n}\boldsymbol{\varphi}_1} = \frac{\sqrt{\left(\sum_j m_{j\ n}x_j\right)^2 + \left(\sum_j m_{j\ n}y_j\right)^2}}{\sum_j\left(m_{j\ n}x_j^2 + m_{j\ n}y_j^2 + I_{j\ n}\theta_j^2\right)}, \tag{5}$$

$$\mathbf{M} = \begin{bmatrix} \mathbf{M}_0 & 0 & 0 \\ 0 & \mathbf{M}_0 & 0 \\ 0 & 0 & \mathbf{I}_0 \end{bmatrix}, \mathbf{M}_0 = \begin{bmatrix} m_1 & & 0 \\ & \ddots & \\ 0 & & m_N \end{bmatrix}, \mathbf{I}_0 = \begin{bmatrix} I_1 & & 0 \\ & \ddots & \\ 0 & & I_N \end{bmatrix}, \tag{6}$$

$$\begin{cases} _\mathbf{n}\mathbf{d} = \left\{\ _nx_1\ \cdots\ _nx_N\quad _ny_1\ \cdots\ _ny_N\quad _n\theta_1\ \cdots\ _n\theta_N\ \right\}^\mathbf{T} \\ _\mathbf{n}\mathbf{f_R} = \left\{\ _nf_{RX1}\ \cdots\ _nf_{RXN}\quad _nf_{RY1}\ \cdots\ _nf_{RYN}\quad _nf_{MZ1}\ \cdots\ _nf_{MZN}\ \right\}^\mathbf{T} \end{cases}, \tag{7}$$

$$_\mathbf{n}\boldsymbol{\varphi}_1 = \left\{\ _n\phi_{X11}\ \cdots\ _n\phi_{XN1}\quad _n\phi_{Y11}\ \cdots\ _n\phi_{YN1}\quad _n\phi_{\Theta11}\ \cdots\ _n\phi_{\Theta N1}\ \right\}^\mathbf{T}, \tag{8}$$

$$_\mathbf{n}\boldsymbol{\alpha_U} = \left\{\ \cos {}_n\psi_1\ \cdots\ \cos {}_n\psi_1\quad -\sin {}_n\psi_1\ \cdots\ -\sin {}_n\psi_1\quad 0\ \cdots\ 0\ \right\}^\mathbf{T}, \tag{9}$$

$$\cos {}_n\psi_1 = \frac{\sum_j m_j \, {}_n\phi_{Xj1}}{\sqrt{\left(\sum_j m_j \, {}_n\phi_{Xj1}\right)^2 + \left(\sum_j m_j \, {}_n\phi_{Yj1}\right)^2}} = \frac{\sum_j m_j \, {}_nx_j}{\sqrt{\left(\sum_j m_j \, {}_nx_j\right)^2 + \left(\sum_j m_j \, {}_ny_j\right)^2}}, \tag{10}$$

$$\sin {}_n\psi_1 = \frac{-\sum_j m_j \, {}_n\phi_{Yj1}}{\sqrt{\left(\sum_j m_j \, {}_n\phi_{Xj1}\right)^2 + \left(\sum_j m_j \, {}_n\phi_{Yj1}\right)^2}} = \frac{-\sum_j m_j \, {}_ny_j}{\sqrt{\left(\sum_j m_j \, {}_nx_j\right)^2 + \left(\sum_j m_j \, {}_ny_j\right)^2}}. \tag{11}$$

In Equations (2)–(11), $m_j$ and $I_j$ are the mass and moment of inertia, respectively, of the $j^{th}$ floor; ${}_nM_{1U}{}^*$ is the equivalent (effective) first modal mass with respect to the $U$-axis at each loading step; and ${}_n\psi_1$ is the angle of incidence of the principal axis ($U$-axis) at each loading step. Figure 2 shows the comparisons of parameters between an $N$-story irregular building and the equivalent SDOF model.

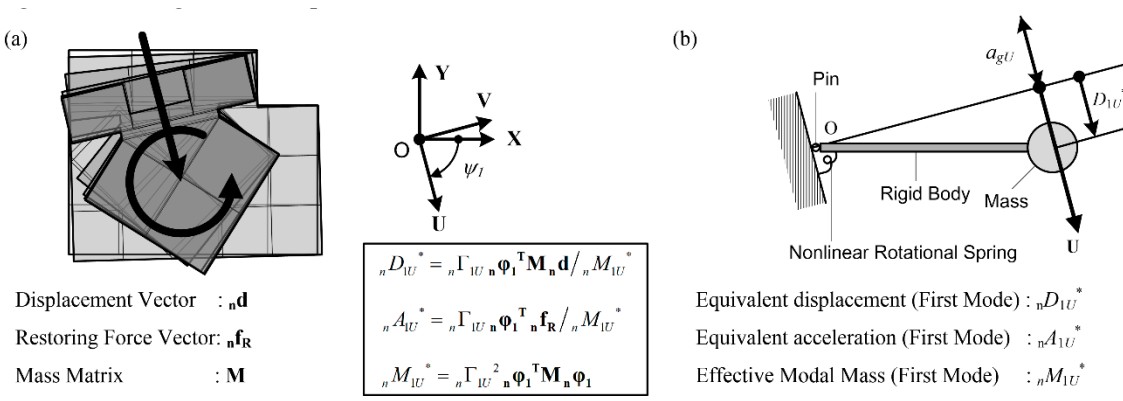

**Figure 2.** Comparisons of parameters between an $N$-story irregular building and the equivalent SDOF model: (**a**) an $N$-story irregular building deformed similar to the first mode shape and (**b**) equivalent SDOF model representing the first modal response.

Pushover analysis was carried out until one of the local responses (e.g., the drift angle of the columns and the plastic hinge rotation angle of the beam) reached its limit value. The displacement limit ($D_{1U}{}^*{}_{limit}$) was obtained as the equivalent displacement corresponding to the loading step, where one of the local responses exceeded its limit value.

2.2.2. Step 2: Calculation of the Scaling Factor Corresponding to Each Pushover Step

The scaling factor corresponding to each pushover step $n$ (${}_n\lambda_1$) is calculated from the given design (or code-specified) response spectrum and the capacity curve obtained in step 1 using the equivalent linearization technique [36]. The equivalent period ${}_nT_{1eq}$ and equivalent damping ratio ${}_nh_{1eq}$ of the equivalent SDOF model at each loading step $n$ are determined as follows:

$$_nT_{1eq} = 2\pi/{}_n\omega_{1eq} = 2\pi\sqrt{{}_nD_{1U}{}^*/{}_nA_{1U}{}^*}, \tag{12}$$

$$_nh_{1eq} = \sum_k \left({}_nh_{eqk}\,{}_nW_{ek}\right)/\sum_k {}_nW_{ek}. \tag{13}$$

In Equation (13), ${}_nh_{eqk}$ and ${}_nW_{ek}$ are the equivalent damping ratio and potential energy, respectively, of the $k^{th}$ nonlinear spring. The equivalent damping ratio of the flexural spring is calculated from Equation (14) with consideration given to the initial damping ratio $h_0$, and the $k^{th}$ ductility of the flexural spring (${}_n\mu_k$).

$$
{}_n h_{eqk} = \begin{cases} h_0 & : {}_n\mu_k < 1 \\ h_0 + 0.25\left(1 - 1/\sqrt{{}_n\mu_k}\right) & : {}_n\mu_k > 1 \end{cases}. \tag{14}
$$

The equivalent damping ratio of the other spring is set to $h_0$ throughout the analysis. In this study, $h_0$ was assumed to be 0.05.

The scaling factor corresponding to each pushover step $n$ $({}_n\lambda_1)$ is calculated as follows:

$$
{}_n\lambda_1 = \frac{{}_nD_{1U}^{*}}{F\left({}_nh_{1eq}\right) \cdot S_D\left({}_nT_{1eq}, 0.05\right)} = \frac{1 + 10{}_nh_{1eq}}{1.5} \cdot \frac{{}_nD_{1U}^{*}}{S_D\left({}_nT_{1eq}, 0.05\right)}. \tag{15}
$$

### 2.2.3. Step 3: Evaluation of the Seismic Capacity Index

The seismic capacity index that considers the unidirectional ground motion ($C_{I, uni}$) was taken from the result obtained by step 2. Additionally, $C_{I, uni}$ is the maximum of ${}_n\lambda_1$ in the range from $D_{1U}^{*} = 0$ to $D_{1U}^{*}{}_{limit}$, as follows:

$$
C_{I,uni} = \max\left\{{}_n\lambda_1\left({}_nD_{1U}^{*}\right); 0 \leq {}_nD_{1U}^{*} \leq D_{1U}^{*}{}_{limit}\right\}. \tag{16}
$$

### 2.3. Evaluation of the Seismic Capacity Index Considering the Bidirectional Ground Motion

#### 2.3.1. Steps 1 to 3: Evaluation of the Seismic Capacity Index Considering the Unidirectional Ground Motion

The seismic capacity index that considers the unidirectional ground motion ($C_{I, uni}$) was evaluated according to the steps described in Section 2.2.

#### 2.3.2. Step 4: Prediction of the Peak Response of the Equivalent SDOF Model (first mode)

The peak response of the equivalent SDOF model representing the first mode (the peak equivalent displacement under unidirectional excitation $D_{1U}^{*}{}_{uni}$, and the equivalent acceleration $A_{1U}^{*}{}_{uni}$ corresponding to $D_{1U}^{*}{}_{uni}$) for scaling, given the ground motion by multiplying $C_{I, uni}$, was obtained using the equivalent linearization technique. Note that $D_{1U}^{*}{}_{uni}$ may be smaller than $D_{1U}^{*}{}_{limit}$. In this case, $C_{I, uni}$ is not a value at $D_{1U}^{*}{}_{limit}$.

#### 2.3.3. Step 5: Pushover Analysis of the Building Model (second mode)

From the results of steps 1 and 4, the first mode vector and the principal axis angle of the first modal response corresponding to $D_{1U}^{*}{}_{uni}$, namely, $\Gamma_{1Uie}\boldsymbol{\varphi}_{1ie}$ and $\psi_{1ie}$, respectively, are obtained. The second mode vector ($\Gamma_{2Vie}\boldsymbol{\varphi}_{2ie}$) is determined from Equations (17) and (18) in terms of $\Gamma_{1Uie}\boldsymbol{\varphi}_{1ie}$ and the second mode vector at the elastic range ($\Gamma_{2Ve}\boldsymbol{\varphi}_{2e}$) with consideration given to the orthogonality of the mode vectors, as follows:

$$
\Gamma_{2Vie} = \frac{\boldsymbol{\varphi}_{2ie}^{\mathbf{T}}\mathbf{M}\boldsymbol{\alpha}_{\mathbf{Vie}}}{\boldsymbol{\varphi}_{2ie}^{\mathbf{T}}\mathbf{M}\boldsymbol{\varphi}_{2ie}}, \tag{17}
$$

$$
\boldsymbol{\varphi}_{2ie} = \boldsymbol{\varphi}_{2e} - \frac{\boldsymbol{\varphi}_{2ie}^{\mathbf{T}}\mathbf{M}\boldsymbol{\varphi}_{1ie}}{\boldsymbol{\varphi}_{1ie}^{\mathbf{T}}\mathbf{M}\boldsymbol{\varphi}_{1ie}}\boldsymbol{\varphi}_{1ie}, \tag{18}
$$

$$
\boldsymbol{\alpha}_{\mathbf{Vie}} = \left\{ \sin\psi_{1ie} \quad \cdots \quad \sin\psi_{1ie} \quad \cos\psi_{1ie} \quad \cdots \quad \cos\psi_{1ie} \quad 0 \quad \cdots \quad 0 \right\}^{\mathbf{T}}. \tag{19}
$$

Next, an additional pushover analysis for an MDOF model was carried out to obtain the force-displacement relationship representing the second mode response by applying the invariant force distribution $\mathbf{p_2}$, which is determined as follows:

$$
\mathbf{p_2} = \mathbf{M}(\Gamma_{2Vie}\boldsymbol{\varphi}_{2ie}). \tag{20}
$$

The equivalent displacement $_nD_{2V}{}^*$ and the equivalent acceleration $_nA_{2V}{}^*$ of the equivalent SDOF model representing the second modal response at each loading step $n$ are determined as follows:

$$_nD_{2V}{}^* = \Gamma_{2Vie}\boldsymbol{\varphi_{2ie}}{}^{\mathbf{T}}\mathbf{M_n d}/M_{2Vie}{}^*, \tag{21}$$

$$_nA_{2V}{}^* = \Gamma_{2Vie}\boldsymbol{\varphi_{2ie}}{}^{\mathbf{T}}\mathbf{_n f_R}/M_{2Vie}{}^*, \tag{22}$$

$$M_{2Vie}{}^* = \Gamma_{2Vie}{}^2\boldsymbol{\varphi_{2ie}}{}^{\mathbf{T}}\mathbf{M}\boldsymbol{\varphi_{2ie}}. \tag{23}$$

In Equation (23), $M_{2Vie}{}^*$ is the equivalent second modal mass with respect to the $V$-axis, and is determined in terms of $\Gamma_{2Vie}\boldsymbol{\varphi_{2ie}}$.

### 2.3.4. Step 6: Peak Response Prediction for the Equivalent SDOF Model (second mode)

The peak response of the equivalent SDOF model representing the second mode ($D_{2V}{}^*{}_{uni}$, which is the peak equivalent displacement under unidirectional excitation, and $A_{2V}{}^*{}_{uni}$, which is the equivalent acceleration corresponding to $D_{2V}{}^*{}_{uni}$) for scaling, given the ground motion by multiplying $C_{I,\,uni}$, was obtained using the equivalent linearization technique.

In the current version of MABPA [26–29], the principal axes of the first and second modes are assumed to be almost orthogonal. Moreover, the principal axis of the second modal response is almost identical to the $V$-axis. As has been discussed in the literature [28,29], this assumption is valid in the case where the torsional indices of the first and second modes, namely, $R_{\rho 1}$ and $R_{\rho 2}$, respectively (defined in Equation (24)) are smaller than 1.

$$R_{\rho i} = \sqrt{\sum_j I_j \phi_{\Theta ji}{}^2 / \sum_j m_j \left(\phi_{Xji}{}^2 + \phi_{Yji}{}^2\right)}. \tag{24}$$

However, this assumption may not be valid if $R_{\rho 1}$ or $R_{\rho 2}$ is larger than 1, as discussed in Fujii [29]. Therefore, in this study, the response spectrum used to predict the peak response ($D_{2V}{}^*{}_{uni}$ and $A_{2V}{}^*{}_{uni}$) was modified by dividing $|\sin\Delta\psi_{12}|$, where $\Delta\psi_{12}$ ($= \psi_1 - \psi_2$) is the angle between the principal axes of the first and second modal responses. Details regarding this modification can be found in Appendix A.

### 2.3.5. Step 7: Pushover Analysis of the Building Model Considering the Bidirectional Seismic Input

In this step, the equivalent displacement limit of the first and the second mode considering the bidirectional seismic inputs $D_{1U}{}^*{}_{bi}$ and $D_{2V}{}^*{}_{bi}$, respectively, are evaluated according to four pushover analyses considering the bidirectional seismic input.

First, from the results obtained in steps 4 and 6, the combined force distributions, namely $\mathbf{p_U}^+$, $\mathbf{p_U}^-$, $\mathbf{p_V}^+$, and $\mathbf{p_V}^-$, are determined as follows:

$$\begin{cases} \mathbf{p_U}^{\pm} = \mathbf{M}(\Gamma_{1Uie}\boldsymbol{\varphi_{1ie}}A_{1U}{}^*{}_{uni} \pm 0.5\Gamma_{2Vie}\boldsymbol{\varphi_{2ie}}A_{2V}{}^*{}_{uni}) \\ \mathbf{p_V}^{\pm} = \mathbf{M}(\pm 0.5\Gamma_{1Uie}\boldsymbol{\varphi_{1ie}}A_{1U}{}^*{}_{uni} + \Gamma_{2Vie}\boldsymbol{\varphi_{2ie}}A_{2V}{}^*{}_{uni}) \end{cases}. \tag{25}$$

Next, pushover analysis (termed as pushover 1U and 2U, respectively) is carried out using $\mathbf{p_U}^+$ or $\mathbf{p_U}^-$ until a) the equivalent displacement $_nD_U{}^*$ calculated from Equation (26) reaches $D_{1U}{}^*{}_{uni}$, as obtained in step 4, or b) one of the local responses reaches its limit value, as follows:

$$_nD_U{}^* = \Gamma_{1Uie}\boldsymbol{\varphi_{1ie}}{}^{\mathbf{T}}\mathbf{M_n d}/M_{1Uie}{}^*, \tag{26}$$

$$M_{1Uie}{}^* = \Gamma_{1Uie}{}^2\boldsymbol{\varphi_{1ie}}{}^{\mathbf{T}}\mathbf{M}\boldsymbol{\varphi_{1ie}}. \tag{27}$$

In Equation (27), $M_{1Uie}{}^*$ is the equivalent first modal mass with respect to the $U$-axis, and is determined in terms of $\Gamma_{1Uie}\boldsymbol{\varphi_{1ie}}$. The displacement limit of the first mode considering the bidirectional

ground motion ($D_{1U}{}^{*}{}_{bi}$) is the minimum value of a) the equivalent displacement $_{n}D_{U}{}^{*}$, when one of the local responses reaches its limit value in pushovers 1U and 2U, and b) the peak equivalent displacement $D_{1U}{}^{*}{}_{uni}$ obtained in step 4.

Similarly, pushover analysis (termed as pushover 1V and 2V, respectively) is performed using $\mathbf{p_V}^{+}$ or $\mathbf{p_V}^{-}$ until a) the equivalent displacement $_{n}D_{V}{}^{*}$ calculated from Equation (28) reaches $D_{2V}{}^{*}{}_{uni}$, as obtained in step 6, and b) one of the local responses reaches its limit value:

$$_{n}D_{V}{}^{*} = \Gamma_{2Vie}\boldsymbol{\varphi_{2ie}}^{\mathbf{T}}\mathbf{M_{n}d}/M_{2Vie}{}^{*}. \tag{28}$$

The displacement limit of the second mode that considers the bidirectional ground motion ($D_{2V}{}^{*}{}_{bi}$) is the minimum value of a) the equivalent displacement $_{n}D_{V}{}^{*}$, when one of the local responses reaches its limit value in pushovers 1V and 2V, and b) the peak equivalent displacement $D_{2V}{}^{*}{}_{uni}$ obtained in step 6.

### 2.3.6. Step 8: Evaluation of the Seismic Capacity Index Considering the Bidirectional Ground Motion

From the results obtained in steps 2 and 7, the scaling factor obtained from the first modal response considering the bidirectional seismic input $\lambda_{1bi}$ is obtained as the maximum $_{n}\lambda_{1}$ in the range from $D_{1U}{}^{*} = 0$ to $D_{1U}{}^{*}{}_{bi}$, as follows:

$$\lambda_{1bi} = \max\left\{_{n}\lambda_{1}\left(_{n}D_{1U}{}^{*}\right); 0 \leq {}_{n}D_{1U}{}^{*} \leq D_{1U}{}^{*}{}_{bi}\right\}. \tag{29}$$

Similarly, the scaling factor obtained from the second modal response considering the bidirectional seismic input $\lambda_{2bi}$ is obtained as the maximum $_{n}\lambda_{2}$ in the range from $D_{2V}{}^{*} = 0$ to $D_{2V}{}^{*}{}_{bi}$, as follows:

$$\lambda_{2bi} = \max\left\{_{n}\lambda_{2}\left(_{n}D_{2V}{}^{*}\right); 0 \leq {}_{n}D_{2V}{}^{*} \leq D_{2V}{}^{*}{}_{bi}\right\}, \tag{30}$$

$$_{n}\lambda_{2} = \frac{1}{\left|\sin \Delta\psi_{12}\right|} \cdot \frac{1 + 10 {}_{n}h_{2eq}}{1.5} \cdot \frac{_{n}D_{2V}{}^{*}}{S_{D}\left(_{n}T_{2eq}, 0.05\right)}. \tag{31}$$

The seismic capacity index considering the bidirectional ground motion ($C_{I, bi}$) is obtained as the minimum of $\lambda_{1bi}$ and $\lambda_{2bi}$, and does not exceed $C_{I, uni}$.

## 3. Building and Ground Motion Data

### 3.1. Basic Information

Figure 3 shows the structural plan of the main Uto City Hall building. The structure of this building can be divided into two structural blocks, namely, the office block and the stair block. The two blocks are connected only by a concrete slab with a thickness of 110 mm. Additionally, as shown in the figure, all of the structural walls are concentrated to the stair block, whereas, in the office block the concrete columns are the only vertical members resisting the lateral loads. Moreover, it should be noted that in the office block, not all frames are oriented in the X- or Y-directions. Specifically, frames $A_1$-$A_3$ lie on the axis rotated 45° counterclockwise from the X-axis (denoted as A-axis), and frames $B_1$-$B_3$ are orthogonal to frames $A_1$ (denoted as B-axis).

Figure 4 shows the simplified structural elevation of frames $Y_4$, $B_1$, and $B_2$. As shown in Figure 4b, the beam for the second-floor slab is doubled in frame $B_1$. The reason for this is that, for levels $Z_1'$ and $Z_1$, the floor level is different in the second floor, as shown in Figure 4.

Figure 5 shows the Uto City Hall after the 2016 Kumamoto Earthquake. As can be seen in the figure, the structural damage observed in frame $B_1$ was more severe than that incurred by frame $A_3$. That is, frame $B_1$ partially collapsed in the fourth story.

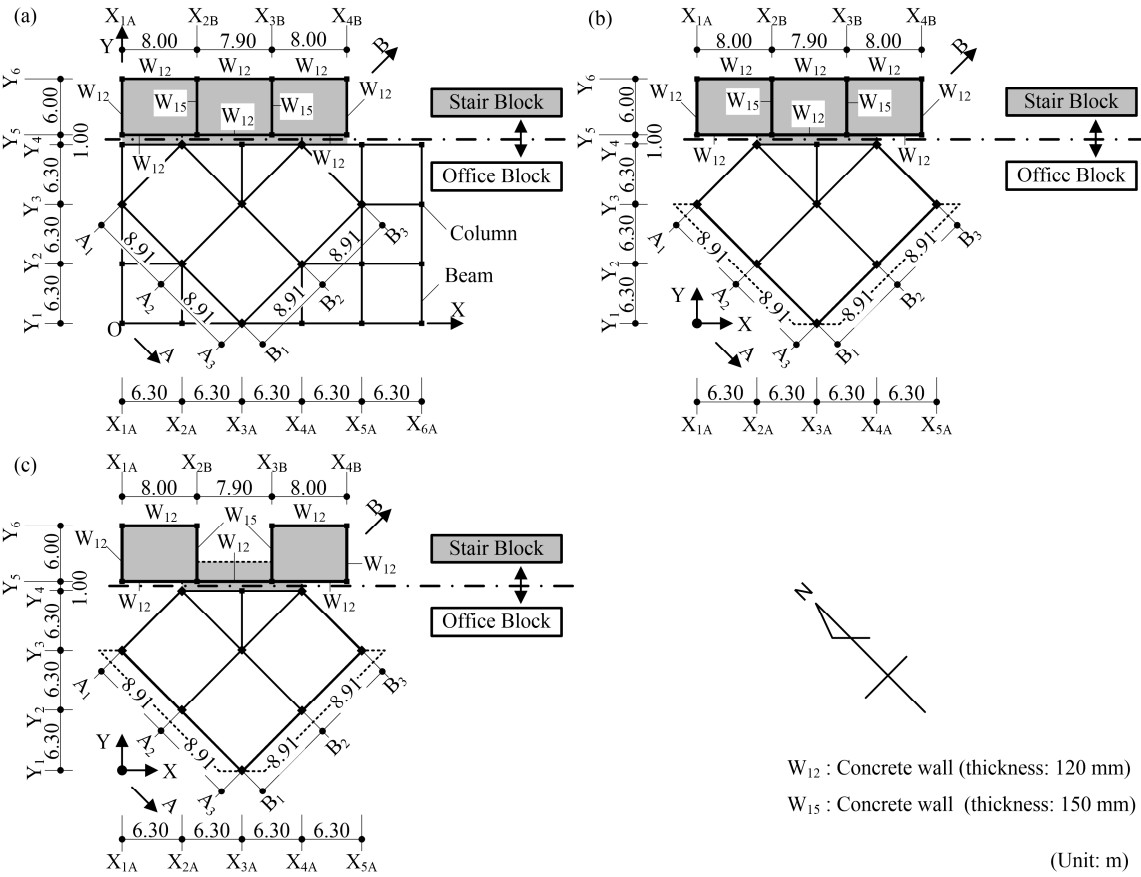

**Figure 3.** Structural plan of the main Uto City Hall building: (**a**) level $Z_0$, (**b**) level $Z_1$ to $Z_4$, and (**c**) level $Z_5$.

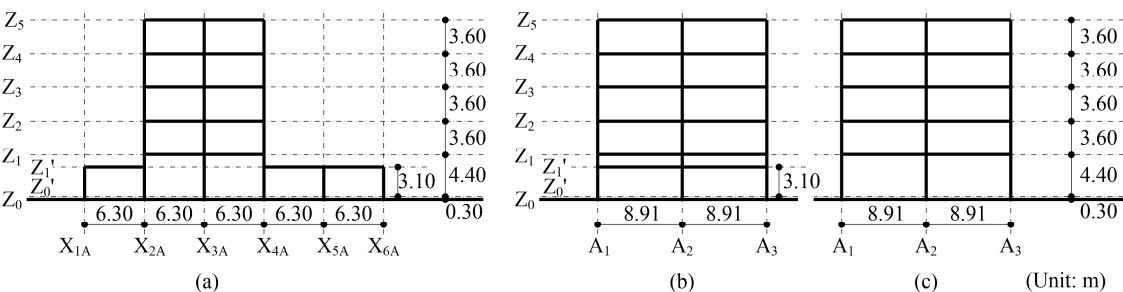

**Figure 4.** Simplified structural elevation of Uto City Hall: (**a**) frame $Y_4$, (**b**) frame $B_1$, and (**c**) frame $B_2$.

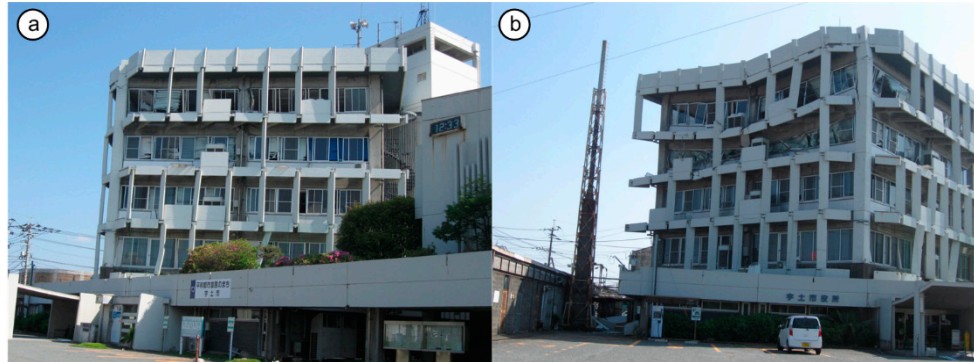

**Figure 5.** View of Uto City Hall after the 2016 Kumamoto Earthquake: (**a**) south (frame $A_3$) and (**b**) southwest (frames $A_3$ and $B_1$).

Figure 6 shows the failure of the beam-column joint observed in frame $B_1$. As is shown in this figure, the part above the column $A_2B_1$ in the third story had slipped down due to the joint failure. Details regarding the structural damages and other information can be found in the literature [1]. Some of the other photos of the structural damages can be found in Appendix B.

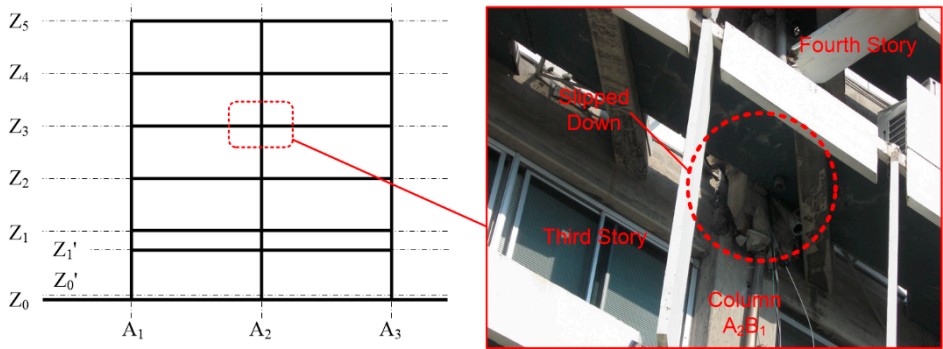

**Figure 6.** Failure of the beam-column joint above the column $A_2B_1$ in the third story.

Figure 7 shows the sections of column $A_1B_1$, $A_2B_1$, and $A_2B_2$. In this building, the deformed bar was used for the longitudinal reinforcement of the beams and the columns, while the round bar was used for the shear reinforcement.

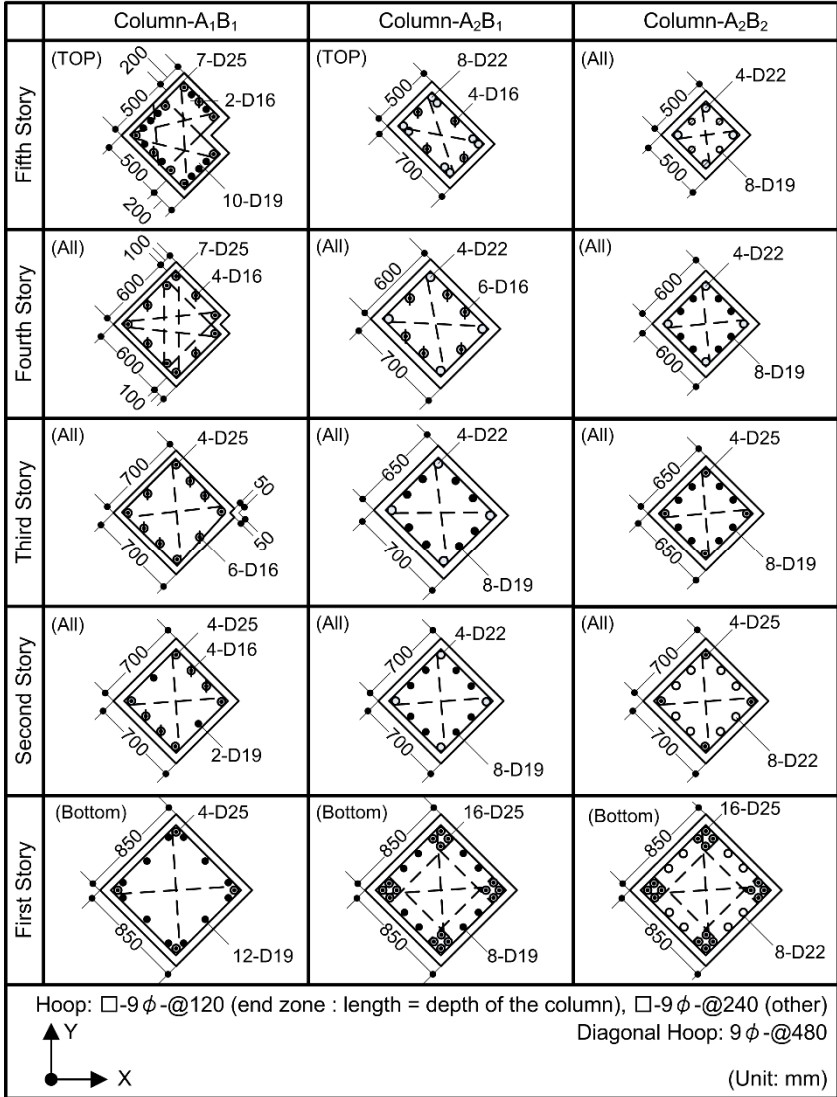

**Figure 7.** Sections of Column $A_1B_1$, $A_2B_1$, and $A_2B_2$ [2].

As shown in this figure, the sectional area of the column is drastically reduced from lower stories to upper stories. This is very common in the reinforced concrete buildings constructed before 1981 because at that time the design seismic force in upper stories is smaller than that in current seismic design code of Japan.

In the previous study [2], the author has made the preliminary seismic evaluation of this building according to the simplified procedure by Shiga in the 1970s, which is based the wall-area index and the average shear stress in walls and columns [37]. The result of the preliminary seismic evaluation is summarized as follows.

Figure 8 shows the evaluation results shown in reference [2]. In this figure, $A_{Wi}$ (unit: mm$^2$) and $A_{Ci}$ (unit: mm$^2$) are the sum of the sectional area of walls and columns in $i$th story, respectively, $A_i$ is the coefficient of the design seismic story shear force used in the current seismic design code of Japan, $A_{fj}$ (unit: m$^2$) is the area of $j$th floor, and $\tau_{avei}$ (unit: N/mm$^2$) is the average shear stress in walls and columns in $i$th story, assuming the weight per unit floor area of the building is 10kN/m$^2$ and base shear coefficient is 1.0. The zone A is the area corresponds to the most of buildings were heavily damaged while the zone C is the area corresponds to the most of buildings were not damaged or only slightly damaged, in 1968 Tokachi-oki Earthquake [37]. Note that in each case, the evaluation is made for the X- and Y-directions, as shown in Figure 3. In Figure 8, the border curve between zones A and B shown in

red is determined by assuming the ultimate shear stress in columns equals 1.2N/mm$^2$ while that in walls equals 3.3 N/mm$^2$. Besides, the horizontal border line between zones B and C is determined as the line the average shear stress $\tau_{avei}$ equals to 1.2 N/mm$^2$, and the vertical border line is determined as the line the wall-area index $A_{Wi} / (A_i \Sigma A_{fj})$ equals to $30 \times 10^2$ mm$^2$/m$^2$. The curves shown in black is the contour of equal column-area index $A_{Ci} / (A_i \Sigma A_{fj})$.

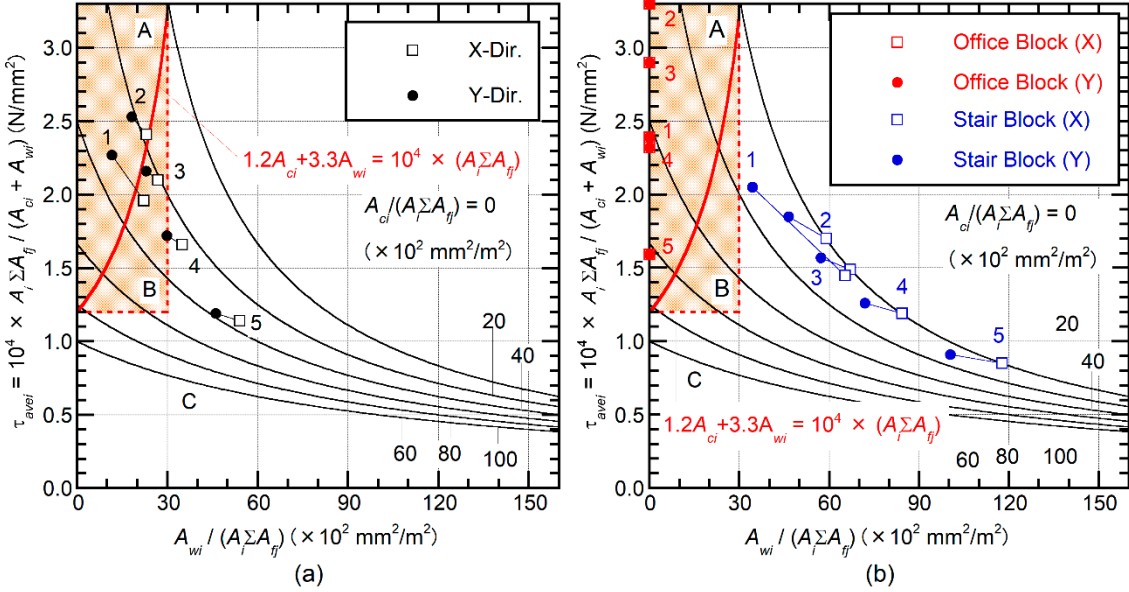

**Figure 8.** Preliminary seismic capacity evaluation results of the main building of the Uto City Hall [2]: (**a**) the case in which the building is assumed to behave as a unite building, and (**b**) the case in which the stair and office blocks are assumed to behave independently.

Figure 8a shows the result of the case in which the building is assumed to behave as a unite building. In this figure, the plots of Y-direction in the first and second story are within zone A, while the plots of upper stories are in zone B or C. Therefore, it may be concluded that the seismic capacity of this building is insufficient to survive strong earthquakes, if this building behaved as a unite single building. However, this result cannot explain the fact that most of the damage in this building is in the upper stories; the most critical of which is the second story in the Y-direction, where the average shear stress is the largest.

Figure 8b shows the results of the case in which the stair and office blocks are assumed to behave independently. In this figure, the plots of office block in all stories are within zone A, while the plots of stair block in all stories are within zone C. Therefore, it may be concluded that the seismic capacity of the office block is insufficient while that of stair block is sufficient, if these two blocks of this building behaved as two independent buildings. However, in the damage observation of this building [1], it is found that the walls in the fifth story of frame $Y_5$ (stair block) are severely damaged. The results shown in Figure 8b cannot explain this damage. Therefore, the assumption that the stair and office blocks behave independently has seemed invalid, even though the floor slab at the border of two blocks in the fourth and fifth floors were severely damaged [1].

From the preliminary evaluation, it may be concluded that the seismic capacity of this building is insufficient to survive strong earthquakes. However, neither results can explain the damage of this building observed. The reasons why this simplified evaluation method fails to explain the observed damage are (i) the lateral force distribution coefficient, $A_i$, is smaller in upper stories because the $A_i$ coefficient cannot reflect the drastic reduction of the sectional area in the upper stories, and (ii) the effect of torsion is not considered in this simplified method. Other reasons would be (iii) the failure of the beams and beam-column joints are not considered in this simplified method.

In the following sections, the discussions are mainly focused on the difference in the damage observed in frame $B_1$ and $A_3$. Because this building had an irregular plan, the effect of torsion would be significant. The author thinks the difference of the damage in these frames may be caused due to the torsional response. Therefore, the three-dimensional nonlinear static and dynamic analyses using spatial frame model were carried out in which the effect of torsion can be considered.

### 3.2. Analysis Model

The main Uto City Hall building is modelled as a three-dimensional spatial frame, wherein the floor diaphragms are assumed to be rigid in their own planes without out-of-plane stiffness. Moreover, in this analysis model, the slabs connecting the office and stair blocks were considered as rigid. The compressive strength of the concrete was assumed 24 N/mm$^2$, while the yield strength of the deformed bar and the round bar were assumed 343 N/mm$^2$ and 294 N/mm$^2$, respectively.

Figure 9 shows the structural model of frames $B_1$ and $Y_5$.

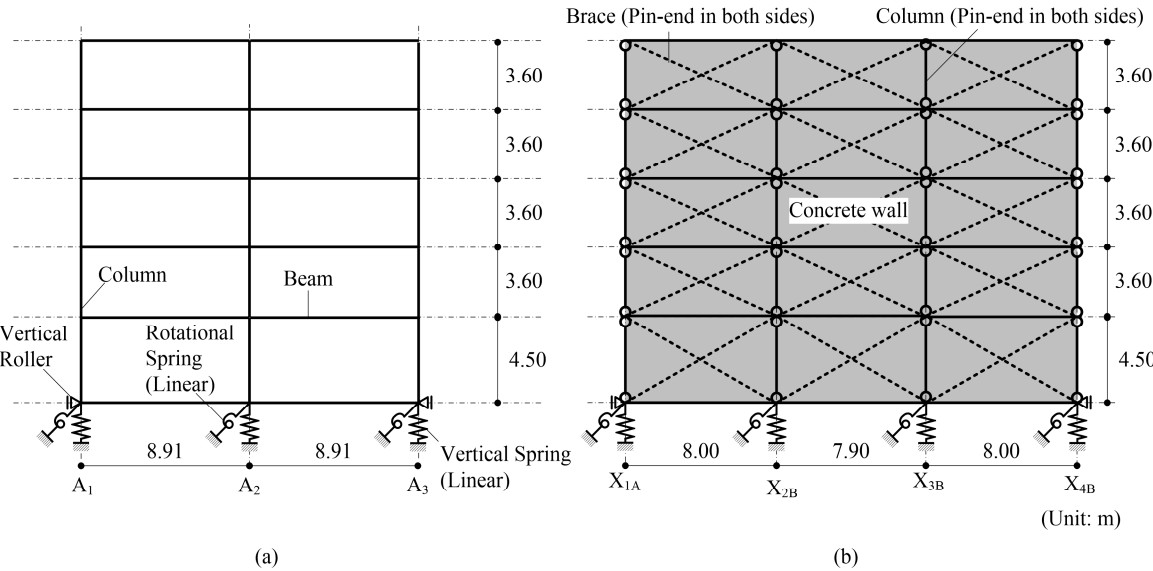

**Figure 9.** Model of each frame: (**a**) frame $B_1$ and (**b**) frame $Y_5$.

As shown in Figure 9a, the double beam in the second-floor level, which is shown in Figure 4b (beams in levels $Z_1$ and $Z_1'$), is modeled as a single beam. The properties of the second-floor beams were determined by considering both beam sections. A one-component model with one non-linear flexural spring at each end and one linear shear spring at the middle of the line element was used for all beams and columns. The bidirectional interaction of the columns' bending moment is not considered because the rigid zone length of some columns is different in each orthogonal direction (e.g., the second story column $A_2B_1(X_{2A}Y_2)$ in frames $A_1$ and $B_2$, and in frames $X_{2A}$ and $Y_2$). The axes for determining the bending moment and rotation were taken as the *A*- and *B*-axis for the columns belonging to the frames oriented in the *A*- and *B*-axis, whereas the rest were taken as the *X*- and *Y*-axis. To determine the flexibility of the springs, we assume an antisymmetric curvature distribution. For the beams (except for the beam within the concrete wall), a rigid zone length is assumed as half the depth of the intersected column minus one-fourth of the depth of the considered beam. For the beams within the concrete wall, the rigid zone length was not considered.

Additionally, the stiffness of the beams within the wall is 100 times higher than that calculated by considering a rectangular beam. For the columns, the rigid zone length is assumed as half the depth of the intersected beam minus one-fourth of the depth of the considered column. The shear behavior of the beams and columns is assumed to be linear elastic. The concrete walls were modeled as two

diagonal braces, as shown in Figure 9b, assuming that the shear behavior is predominant. The axial behavior of all vertical members is assumed to be linear elastic.

Figure 10 shows the force-deformation relationship of the nonlinear flexural spring.

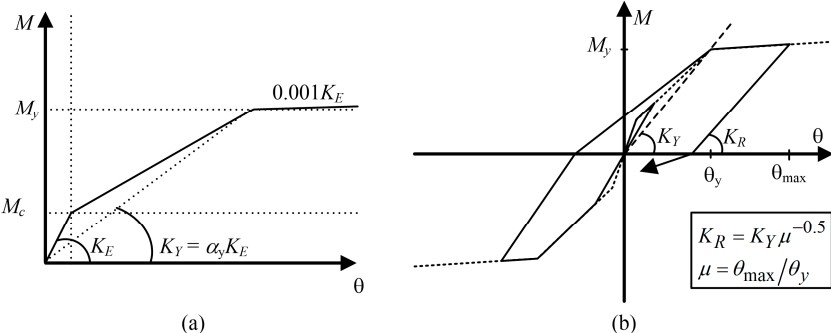

**Figure 10.** Force-deformation relationship of nonlinear flexural spring: (**a**) envelope and (**b**) hysteresis rule.

The envelopes are assumed to be symmetric in the positive and negative loading directions. The yield moment $M_y$ of each member is calculated according to the AIJ standard [38], whereas the crack moment $M_c$ is assumed to be one-third of $M_y$. For simplicity, only the axial force attributed to the vertical load was considered in the calculation of each column's yield moment. Moreover, as has been described in a previous report, various beam-column joint failures were observed [1]. Additionally, the yielding moment of the corresponding beam ends was reduced according to the AIJ standard [38] to consider the effect of the beam-column joint yield at the ultimate strength. The range of the reduction factor was 0.837 to 1.000. In Figure 10a, the secant stiffness degradation ratio ($\alpha_y$) of the flexural spring at the yield point is calculated according to the equation of Sugano's equation and Koreishi [39]. The tangent stiffness degradation ratio ($\alpha_2$) of the flexural spring beyond the yielding point is assumed to be 0.001 for all beams and columns.

The Muto hysteresis model [40] with one modification was used to model the flexural springs, as shown in Figure 10b. Specifically, the unloading stiffness after yielding was proportionally decreased to $\mu^{-0.5}$ ($\mu$ is the ductility of the flexural spring) to represent the degradation of the unloading stiffness after the yielding of the RC members, in the same way as in Otani's model [41].

Figure 11 shows the simplified concrete wall model. The properties of the diagonal braces were determined to represent the shear force-deformation behavior. The property of the vertical elements (columns at each border and within the walls) was determined to represent the bending and axial stiffness of the concrete wall, and their behavior is assumed to be linear elastic. Figure 11b,c show the force-deformation relationship of the diagonal braces. The envelope shown in Figure 11b is assumed to be symmetric in the positive and negative loading directions. The stiffness and strength of the diagonal braces were determined with consideration given to the existing opening width. The ultimate shear strength of the concrete wall ($Q_{suW}$) is calculated according to the AIJ standard [38] and reduced by considering the ratio of the sum of the opening width divided by the wall length ($l_W$). To determine the axial deformation corresponding to the point of maximum shear strength, we assume the shear strain $\gamma_{su}$ to be 0.004. Similarly, to determine the shear strength's point of loss, we assume the shear strain $\gamma_u$ to be 0.010. The origin-oriented model (Figure 11c) was used to model the non-linear behavior of the diagonal braces.

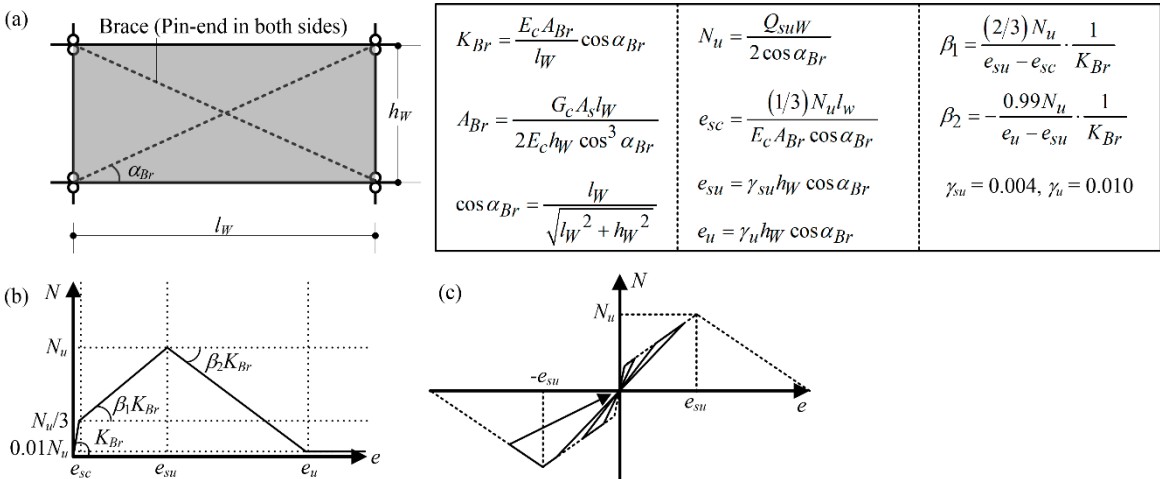

**Figure 11.** Simplified model of the concrete wall: (**a**) model, (**b**) envelope of the force-deformation relationship of the nonlinear brace (concrete wall), and (**c**) hysteresis rule.

The properties of the vertical and rotational springs at each basement are calculated on the bases of the axial stiffness of the piles according to the method presented in FEMA 273 [7]. The vertical and rotational springs are assumed to be linear elastic.

Second-order effects (e.g., the P-Δ effect) were not considered. The damping matrix is assumed to be proportional to the tangent stiffness matrix, with 5% of the first mode's critical damping. Note that, when its tangent stiffness is negative, non-zero dummy positive value ($10^{-9}$ times smaller than the initial stiffness) is used for the calculation of the damping matrix to avoid the negative damping.

The building model developed in this study may not sufficiently represent the behavior of the actual building during the 2016 Kumamoto Earthquake. In particular, the assumption that the behavior of all nonlinear springs, including the diagonal braces representing the behavior of the concrete wall, can be determined as strictly symmetric in the positive and negative loading directions, including the axial deformation of the vertical members (all of them are assumed to have linear elastic behavior) is the most critical consideration. Owing to the nature of the concrete's behavior, the symmetric behavior of the members in the positive and negative loading directions may not be true. However, in this study, this modeling scheme was applied to maintain the stability of the DB-MAP analysis as much as possible after the strength of the wall degraded. For the same reason, the vertical and rotational springs at the basement are assumed to have a linear elastic behavior. Influence of the strength degradation behavior of the concrete walls and stiffness of the vertical and rotational springs at the basement to the whole behavior of the building would be discussed in Section 5.

Figure 12 shows the natural modes of the building model in the elastic range. Here, $T_{ie}$ is the $i$th natural period in the elastic range ($i = 1$–3), $m_{ie}{}^{*}$ is the equivalent (effective) modal mass ratio of the $i$th mode with respect to its principal direction in the elastic range, $\psi_{ie}$ is the incidence angle of the principal direction of the $i$th modal response in the elastic range, and $R_{\rho ie}$ is the torsional index of the $i$th mode in the elastic range.

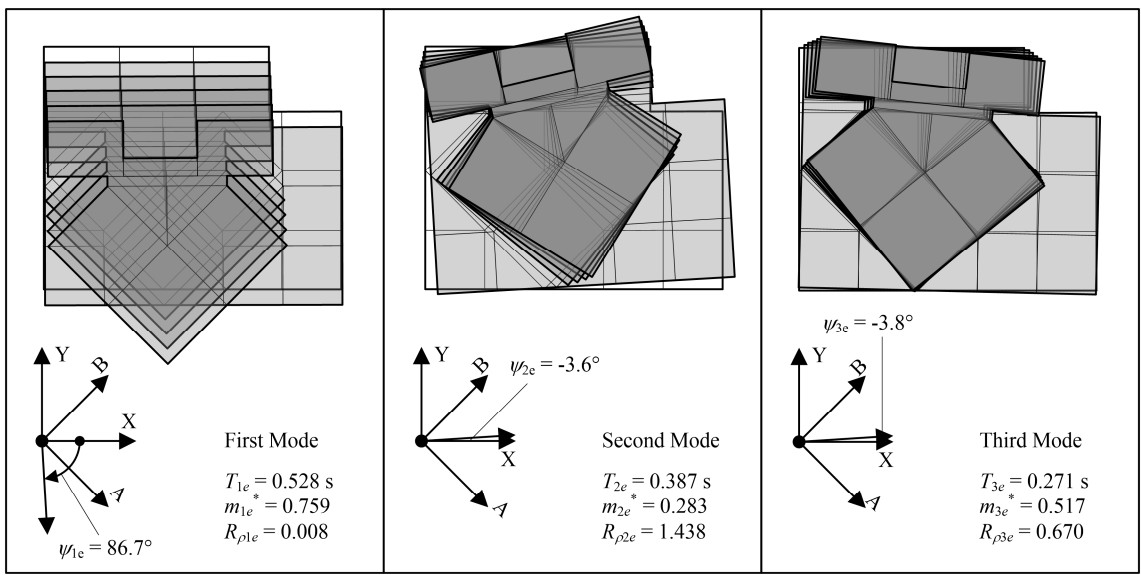

**Figure 12.** The shape of the first three natural modes of the building model in the elastic range.

As shown in the figure, the first mode is almost purely translational ($R_{\rho 1e} = 0.008 \ll 1$), the second mode is predominantly torsional ($R_{\rho 2e} = 1.438 > 1$), and the third mode is predominantly translational ($R_{\rho 3e} = 0.670 < 1$). Therefore, this building cannot be classified as torsionally stiff (TS). However, the equivalent first modal mass ratio is larger than 0.5 ($m_{1e}^* = 0.759 > 0.5$). Therefore, this building may oscillate predominantly in the first mode from the *U*-directional (unidirectional) excitation. Additionally, the angles between the principal directions of the first two modes are close to 90° such that $\Delta \psi_{12} = 86.7° - (-3.6°) = 90.3°$.

### 3.3. Ground Motion Data

In this study, the seismic excitation was bidirectional in the *X-Y* plane, and ten sets of artificial ground motions were generated. Additionally, the response spectra of the major and minor components were assumed to be identical. The target elastic spectra of the major and minor components with 5% critical damping ($_pS_{A\xi}(T, 0.05)$ and $_pS_{A\zeta}(T, 0.05)$, respectively) were determined using the current Building Standard Law of Japan with regard to extremely rare earthquake events and with consideration given to the regional seismic event factor in Uto City ($Z = 0.8$). Moreover, the type-2 soil (normal) is calculated using Equation (32), where *T* represents the natural period of the SDOF model, as follows:

$$_pS_{A\xi}(T, 0.05) = {_pS_{A\zeta}}(T, 0.05) = \begin{cases} 0.8(4.8 + 45T) & \text{m/s}^2 & : T \le 0.16\text{ s} \\ 9.60 & \text{m/s}^2 & : 0.16\text{ s} < T \le 0.864\text{ s} \\ 9.60(0.864/T) & \text{m/s}^2 & : T > 0.864\text{ s} \end{cases} \quad (32)$$

The phase angle is given by the uniform random value and the Jenning-type envelope function *e*(*t*) recommended by the Building Center of Japan [42,43], as follows:

$$e(t) = \begin{cases} (t/5) & : 0\text{ s} < t \le 5\text{ s} \\ 1 & : 5\text{ s} < t \le 35\text{ s} \\ \exp\{-0.027(t - 35)\} & : 35\text{ s} < t \le 120\text{ s} \end{cases} \quad (33)$$

Figure 13 shows the elastic response spectra of the artificial ground motions with 5% critical damping. Note that the artificial ground motions considered in this study were generated independently, i.e., there is no correlation between each component. The correlation coefficients of all ten sets are close to zero, and the envelope functions of the two components are identical. Therefore, the two components can be considered to be independent of each other.

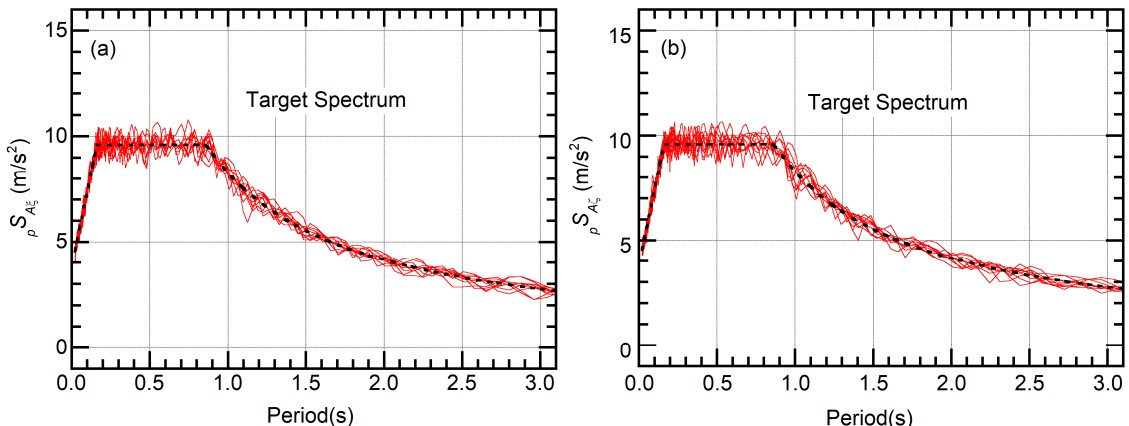

**Figure 13.** Response spectra of the elastic acceleration.

Figure 14 shows an example of accelerations and orbit of the artificial ground motion used in the time-history analysis shown in Section 4.

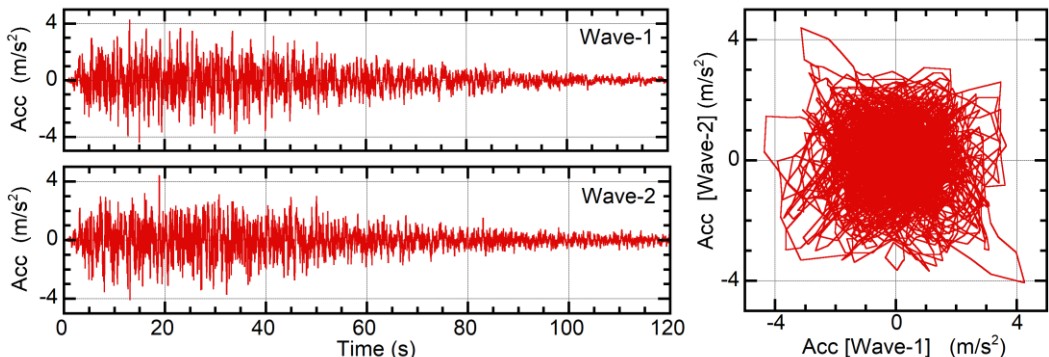

**Figure 14.** Example of accelerations and orbit of the artificial ground motion.

## 4. Validation of the Seismic Capacity Evaluation Procedure

This section presents the seismic capacity evaluation results, obtained by pushover analysis, and their validation. In this study, the displacement limit was determined based on the story drift at each column (vector value) in frames $A_1$ to $A_3$ (and also $B_1$ to $B_3$). The limit value of the story drift was assumed as 1/75 of its height. The reasons for this are that (i) no severe shear failure were observed in all columns, and (ii) the yielding of the beam-column joint, which was observed in frames $A_3$ and $B_1$, affects the collapse mechanism significantly when a larger drift occurs.

### 4.1. Pushover Analysis Results

Figure 15 shows the non-linear properties of the equivalent SDOF model representing the first mode response obtained from the pushover analysis results. As discussed above, the pushover analysis terminated when the story drift angle at column $A_1B_1$ in the second story exceeded 1/75. From Figure 15a, the displacement limit was obtained as $D_{1U}{}^*{}_{limit} = 8.420 \times 10^{-2}$ m. The equivalent period $T_{1eq}$ at $D_{1U}{}^*{}_{limit}$ was 1.191 s, as shown in Figure 15b. The equivalent damping $h_{1eq}$ at $D_{1U}{}^*{}_{limit}$ was $6.115 \times 10^{-2}$, as shown in Figure 15c.

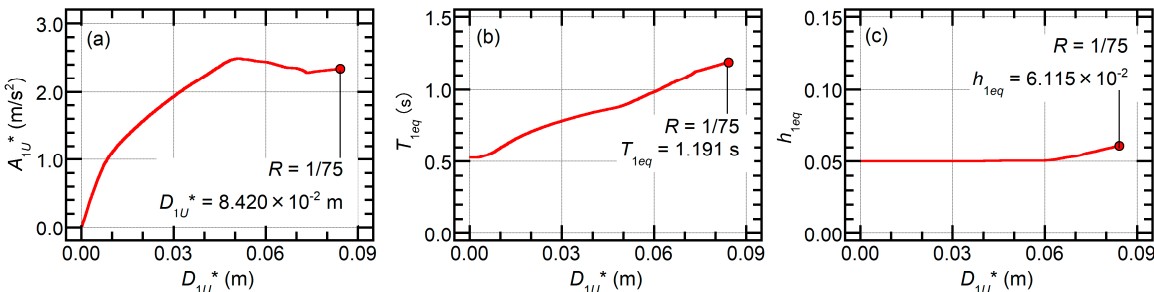

**Figure 15.** Nonlinear properties of the equivalent SDOF model representing the first mode response obtained from the pushover analysis: (**a**) $A_{1U}^*$-$D_{1U}^*$ relationship, (**b**) $T_{1eq}$-$D_{1U}^*$ relationship, and (**c**) $h_{1eq}$-$D_{1U}^*$ relationship.

Figure 16 shows the variations in the first mode parameters (the effective modal mass ratio $m_{1U}^*$, the incidence angle of the principal axis for the first modal response $\psi_1$, and the torsional index of the first mode $R_{\rho1}$) based on the results of the pushover analysis. As shown in Figure 16a, the effective modal mass ratio ($m_{1U}^*$) decreased as the equivalent displacement ($D_{1U}^*$) increased from $m_{1U}^* = 0.759$, in the elastic range, to $m_{1U}^* = 0.5850$ at $D_{1U}^* = D_{1U}^*{}_{limit}$. The incidence angle of the principal axis of the first mode ($\psi_1$) changed from 86.7° in the elastic range to 75.1° at $D_{1U}^* = D_{1U}^*{}_{limit}$, as shown in Figure 16b. The torsional index of the first mode $R_{\rho1}$ increased significantly from 0.008 to 0.516 at $D_{1U}^* = D_{1U}^*{}_{limit}$, as shown in Figure 16c.

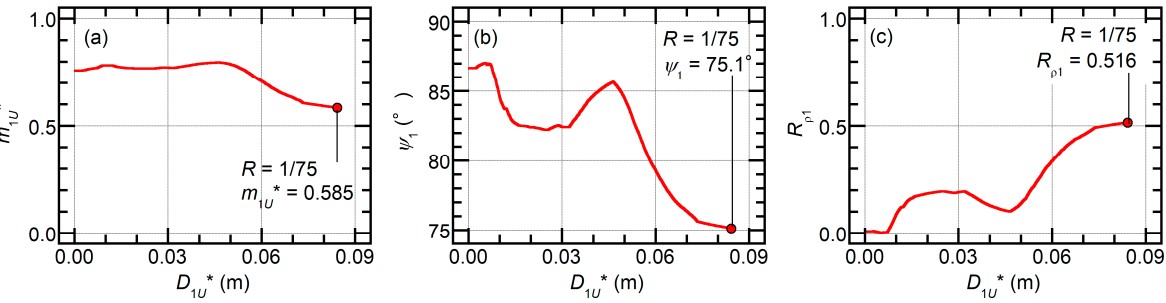

**Figure 16.** Variations in the first mode parameters based on the pushover analysis: (**a**) effective modal mass ratio ($m_{1U}^*$), (**b**) incidence angle of the principal axis of the first modal response ($\psi_1$), and (**c**) torsional index of the first mode ($R_{\rho1}$).

Figure 17 shows the roof displacement responses based on the results of the pushover analysis.

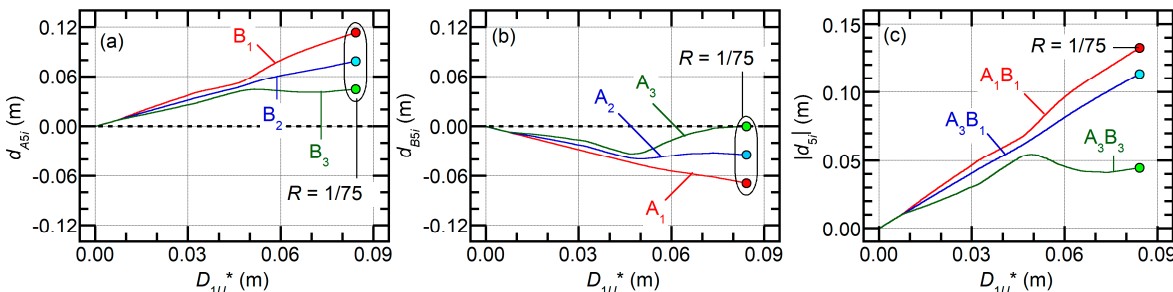

**Figure 17.** Roof displacement responses based on the pushover analysis: (**a**) frames $B_1$ to $B_3$, (**b**) frames $A_1$ to $A_3$, and (**c**) columns $A_1B_1$, $A_3B_1$, and $A_3B_3$.

As shown in Figure 17a, the roof displacement at frame $B_1$ was the largest of the three frames parallel to the *A*-axis throughout the response. Similarly, the displacement at frame $A_1$ was the largest of the three frames parallel to the *B*-axis, as shown in Figure 17b. Therefore, in comparison with the

responses of the three corner columns ($A_1B_1$, $A_3B_1$, and $A_3B_3$), the displacement of column $A_1B_1$ was the largest (Figure 17c).

Figure 18 shows the shape of the first natural mode at $D_{1U}^* = D_{1U}^*{}_{limit}$. In this figure, significant torsion can be observed in the first mode. Here, frame $B_1$ is the most critical (largest deformation observed), while frame $A_3$ is less critical.

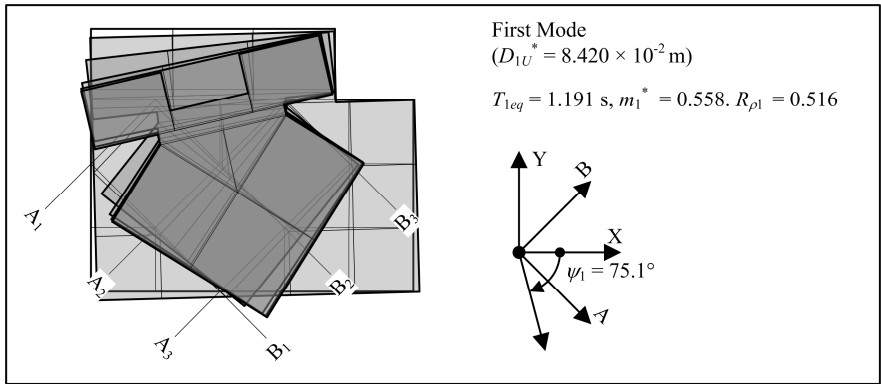

**Figure 18.** The shape of the first natural mode at $D_{1U}^* = 8.420 \times 10^{-2}$ m.

## 4.2. Evaluation of the Capacity Index

First, the capacity index that considers the unidirectional ground motion ($C_{I,\,uni}$) was evaluated. Figure 19 shows the capacity index evaluation result. Figure 19a shows that the evaluated peak response for each scaling factor $\lambda$. From this figure, the relation between the scaling factor corresponding to each pushover step $n$ ($_n\lambda_1$) and equivalent displacement $_nD_{1U}^*$ can be constructed as shown in Figure 19b. Therefore, the estimated value was the value at $D_{1U}^* = D_{1U}^*{}_{limit} = 8.420 \times 10^{-2}$ m and $C_{I,\,uni} = 0.361$.

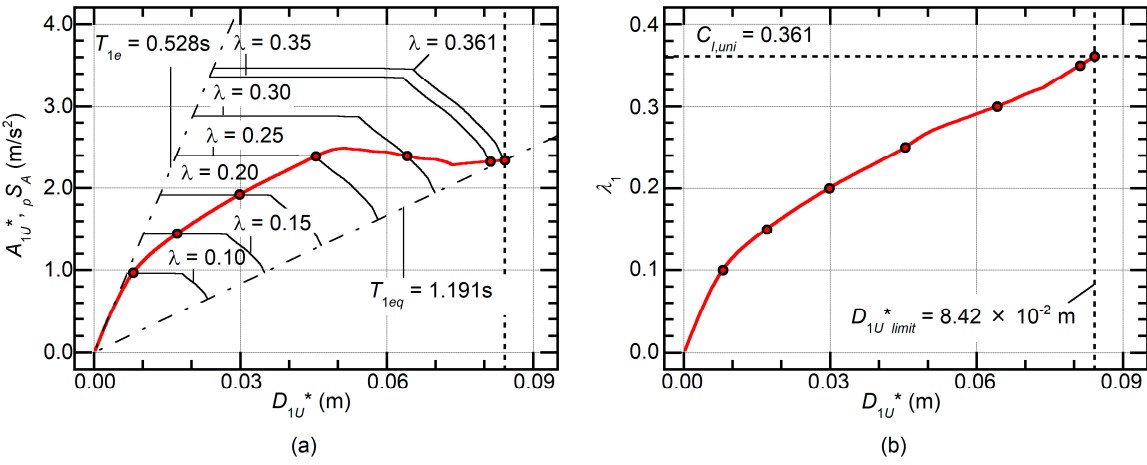

**Figure 19.** Evaluation of the capacity index considering the unidirectional ground motion $C_{I,\,uni}$: (**a**) the evaluation of the peak response for each scaling factor $\lambda$, (**b**) variation of the evaluated scaling factor $\lambda_1$ corresponding to the equivalent displacement $D_{1U}^*$.

Next, the capacity index that considers the bidirectional ground motion ($C_{I,\,bi}$) was evaluated.

Figure 20 shows the peak response prediction for the first and second modal responses of the scaling factor $\lambda = C_{I,\,uni} = 0.361$. Because the angle between the principal axes of the first and second modal responses ($\Delta\psi_{12}$) was not close to 90°, the demand curve for estimating the second mode was factored by $1/|\sin\Delta\psi_{12}|$. Thus, the calculated value of $\Delta\psi_{12}$ is 116.2° ($\psi_1 = 75.1°$ and $\psi_2 = -41.1°$). Thereby, the value of $|\sin\Delta\psi_{12}|$ is 0.8973. Hence, the demand curve was factored by $1/0.8973 = 1.115$. As shown in this figure, the estimated peak responses are $D_{1U}^* = D_{1U}^*{}_{limit} = 8.420 \times 10^{-2}$ m and $D_{2V}^*{}_{uni} = 6.780 \times 10^{-2}$ m, respectively.

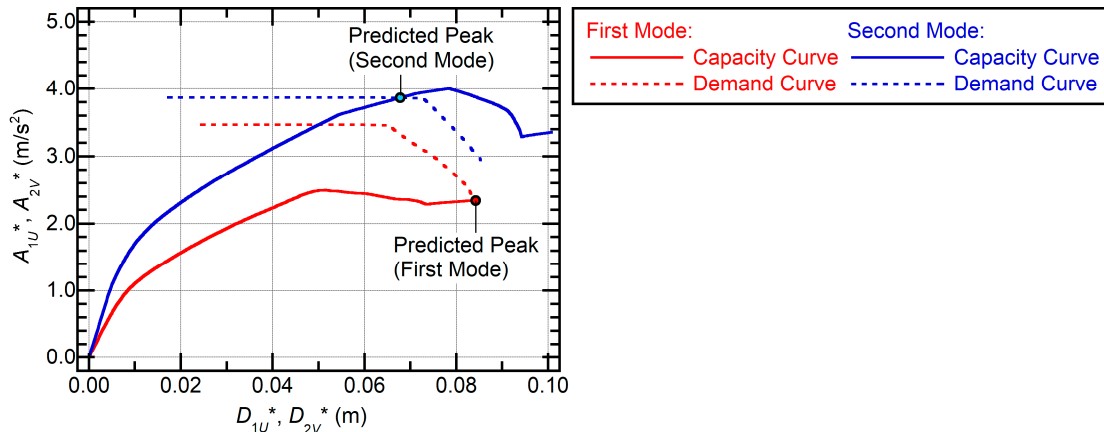

**Figure 20.** Prediction of the peak response for the first and second modal responses ($\lambda = C_{I,\,uni} = 0.361$).

Figure 21 shows the evaluation of the capacity index that considers the bidirectional ground motion. Figure 21a shows the plots of the roof displacement response relationships at column $A_1B_1$, as obtained from the results of steps 1 (first mode) and 7 (pushovers 1U and 2U using the invariant force vector $\mathbf{P_U}^+$ and $\mathbf{P_U}^-$, respectively). As can be seen, the smallest equivalent displacement, where the story drift exceeded 1/75 is $7.54 \times 10^{-2}$ m, and was obtained from the result of pushover 1U. Therefore, the displacement limit of the first mode considering the bidirectional ground motion $D_{1U}{}^*_{bi}$ is $7.54 \times 10^{-2}$ m. From the results of pushovers 1V and 2V (not presented in this paper), the displacement limit of the second mode considering the bidirectional ground motion ($D_{2V}{}^*_{bi}$) is equal to $D_{2V}{}^*_{uni} = 6.780 \times 10^{-2}$ m, because the story drifts did not exceed 1/75 in the results of pushovers 1V and 2V.

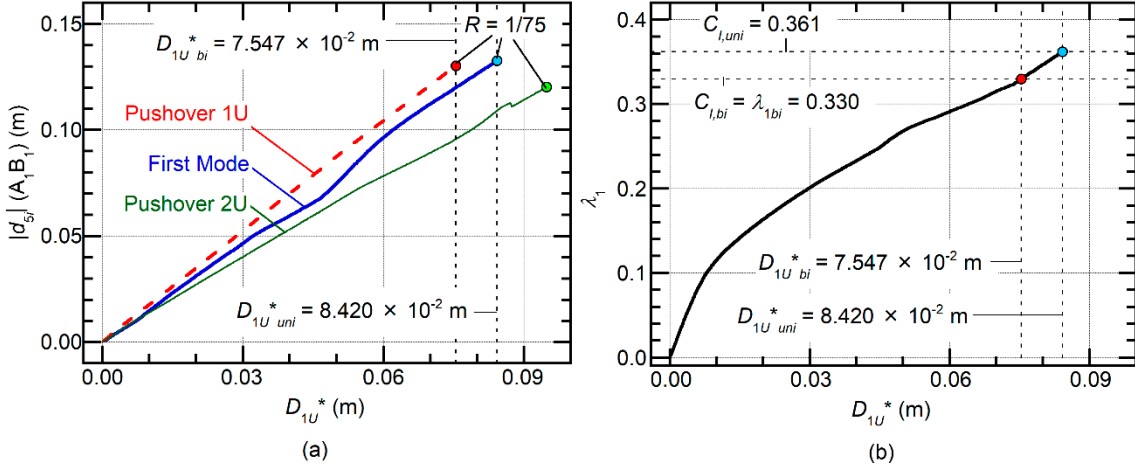

**Figure 21.** Evaluation of the capacity index considering the bidirectional ground motion $C_{I,\,bi}$: (**a**) roof displacement responses at column $A_1B_1$ and (**b**) variation of the scale factor $\lambda$.

Figure 21b shows the variation of the scale factor $\lambda_1$ corresponding to each $D_{1U}{}^*$. The evaluated $\lambda_{1bi}$ corresponding to $D_{1U}{}^*_{bi} = 7.55 \times 10^{-2}$ m is equal to 0.330. In the second mode, because $D_{2V}{}^*_{bi}$ is equal to $D_{2V}{}^*_{uni}$, the evaluated $\lambda_{2bi}$ is equal to $C_{I,\,uni} = 0.361$. Therefore, the evaluated capacity index that considers the bidirectional excitation is the minimum of $\lambda_{1bi} = 0.330$ and $\lambda_{2bi} = 0.361$. Thus, the evaluated $C_{I,\,bi}$ is equal to 0.330.

### 4.3. Validation of the Evaluation Procedure by Nonlinear Time-history Analysis

#### 4.3.1. Analysis Cases

The validity of the procedure was evaluated as follows. Nonlinear time-history analyses were carried out using the 10 sets of artificial ground motions presented in Section 3.3. The peak story drifts at columns $A_3B_3$, $A_1B_1$, and $A_3B_1$ were compared with their drift limit ($R = 1/75$). In the first case, the scale factor of the ground motions ($\lambda$) was taken as the capacity index considering the unidirectional ground motion, which was evaluated above ($\lambda = C_{I, uni} = 0.361$). In the second case, $\lambda$ was taken as the capacity index that considers the bidirectional ground motion ($\lambda = C_{I, bi} = 0.330$). Note that, in both cases, the incidence angle of the major component with respect to the X-axis ($\psi$) was taken as 0°, 45°, 90°, and 135°. Therefore, $10 \times 4 = 40$ analyses were carried out for each case.

In this study, the peak story drifts at columns $A_3B_3$, $A_1B_1$, and $A_3B_1$ were predicted by MABPA [26] with the following modification: the response spectrum used to predict the peak response of the second mode ($D_{2V}{}^*{}_{uni}$ and $A_{2V}{}^*{}_{uni}$) was modified by dividing $|\sin\Delta\psi_{12}|$, as discussed in Section 2.3.4.

#### 4.3.2. Analysis Results

Figures 22 and 23 show the peak drift of columns $A_1B_1$, $A_3B_1$, and $A_3B_3$, as obtained by the time-history analyses for the cases of $\lambda = C_{I, uni} = 0.361$ and $\lambda = C_{I, bi} = 0.330$, respectively. In these figures, the predicted peak drift by MABPA is also shown.

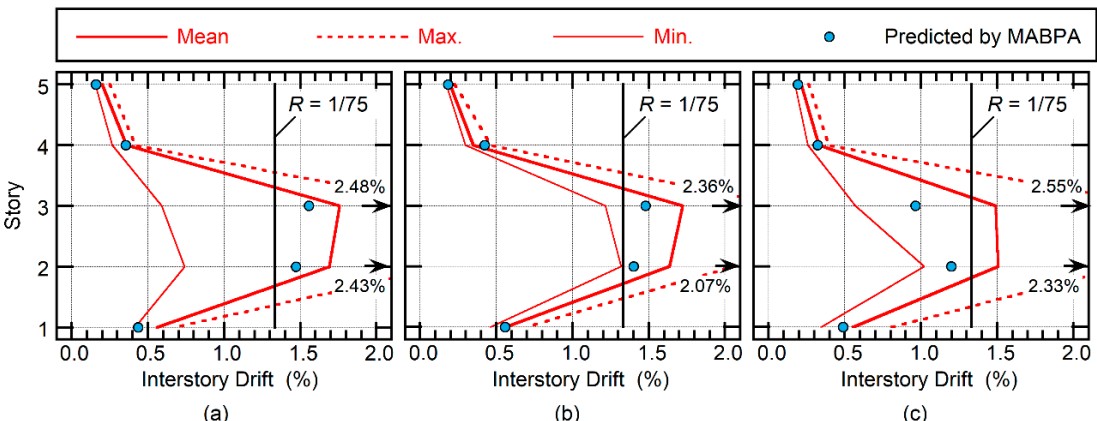

**Figure 22.** Comparison of the peak drift for columns ($\lambda = C_{I, uni} = 0.361$): (**a**) column $A_1B_1$, (**b**) column $A_3B_1$, and (**c**) column $A_3B_3$.

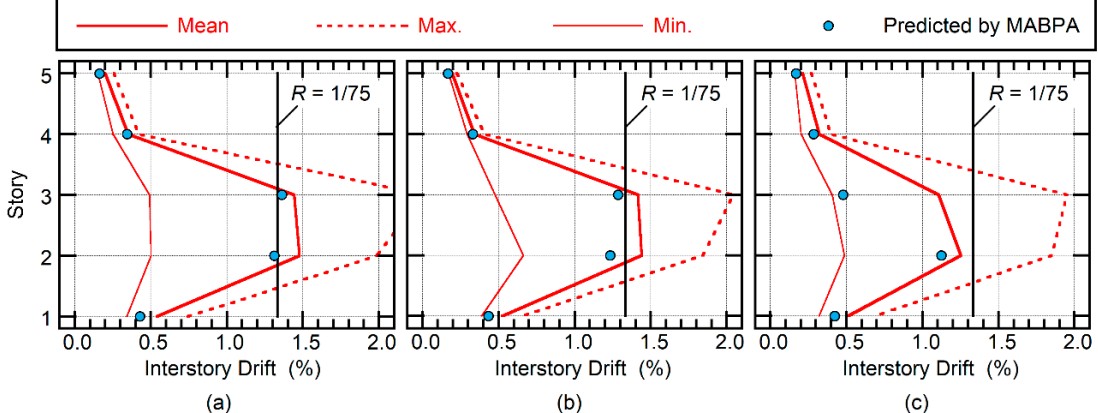

**Figure 23.** Comparison of the peak drift for columns ($\lambda = C_{I, bi} = 0.330$): (**a**) column $A_1B_1$, (**b**) column $A_3B_1$, and (**c**) column $A_3B_3$.

When $\lambda = C_{I,\,uni} = 0.361$, the mean value of the peak drift at all three columns notably exceeded its drift limit ($R = 1/75$), as shown in Figure 22. In addition, the predicted peak drift by MABPA exceeds its drift limit ($R = 1/75$) at column $A_1B_1$ and $A_3B_1$. While in the case of $\lambda = C_{I,\,bi} = 0.330$, the mean peak drift at column $A_1B_1$ and $A_3B_1$ was closer to the drift limit than the case of $\lambda = C_{I,\,uni} = 0.361$, as shown in Figure 23. In addition, the predicted peak drift by MABPA is very close to $R = 1/75$ at column $A_1B_1$.

Therefore, the accuracy of the evaluated seismic capacity index for this building was satisfactory. Moreover, index $C_{I,\,bi}$ was better than index $C_{I,\,uni}$ because in this building, the effect of bidirectional excitation was significant. In addition, note that the accuracy of the predicted peak drift by MABPA for this building was satisfactory.

## 5. Comparisons of the Evaluated Seismic Capacity of the Main Uto City Hall Building and the Response Spectrum of Recorded Earthquakes

In this section, the seismic capacity of the main Uto City Hall building is evaluated, considering the variations of modeling of strength degradation behavior and basement spring. Then the evaluated seismic capacity curves are compared to the response spectrum of the first and second earthquakes. For the evaluation of the seismic capacity of each model, the pushover-based method described in Section 2 is applied, considering the code-specific spectrum shown in Section 3.3.

### 5.1. Analysis Cases

The parameters considered in this section were (i) the force-deformation of the diagonal braces for concrete wall, and (ii) the stiffness of the vertical and rotational springs at each basement, as shown in Figure 24. We assume the shear strain $\gamma_u$ to be 0.01, 0.02, and 0.04, as shown in Figure 24a. In each case, the stiffness of springs at each basement was taken as 100%, and 50% of the calculated values described in Section 3.2, as shown in Figure 24b. Table 1 shows the list of 6 structural models for the parametric study. Note that Model RuW1-100 is the model analyzed for the validation of the evaluation procedure in the previous section.

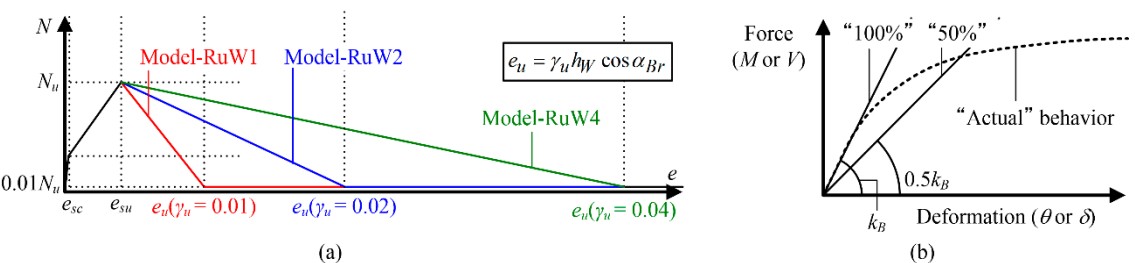

**Figure 24.** Parameters considered in modeling: (**a**) the force-deformation of the diagonal braces for concrete wall, and (**b**) the stiffness of the vertical and rotational springs at each basement.

**Table 1.** List of structural models considered in the parametric study.

| Model ID | Shear Strain $\gamma_u$ | Stiffness Ratio of Spring at Each Basement |
|---|---|---|
| Model-RuW1-100 | 0.01 | |
| Model-RuW2-100 | 0.02 | 100% |
| Model-RuW4-100 | 0.04 | |
| Model-RuW1-050 | 0.01 | |
| Model-RuW2-050 | 0.02 | 50% |
| Model-RuW4-050 | 0.04 | |

### 5.2. Analysis Results

5.2.1. Capacity Curves and Seismic Capacity Indices

Table 2 summarizes the evaluation results of each model. As is shown in this table, the range of evaluated $C_{I, bi}$ is from 0.324 to 0.378. This table also shows that the displacement limit is reduced by about 10% to 15% by considering the bidirectional excitation, while the reduction of the capacity index is about 7% to 11%.

**Table 2.** Summary of the evaluation results of each model.

| Model ID | Displacement Limit | | | Capacity Index | | |
|---|---|---|---|---|---|---|
| | $D_{1U}{}^{*}{}_{uni}$ $(\times 10^{-2}$ m) | $D_{1U}{}^{*}{}_{bi}$ $(\times 10^{-2}$ m) | $D_{1U}{}^{*}{}_{bi}/D_{1U}{}^{*}{}_{uni}$ | $C_{I, uni}$ | $C_{I, bi}$ | $C_{I, bi}/C_{I, uni}$ |
| RuW1-100 | 8.420 | 7.547 | 0.896 | 0.361 | 0.330 | 0.914 |
| RuW2-100 | 8.760 | 7.880 | 0.900 | 0.393 | 0.365 | 0.929 |
| RuW4-100 | 9.025 | 8.045 | 0.891 | 0.412 | 0.378 | 0.917 |
| RuW1-050 | 8.616 | 7.480 | 0.868 | 0.356 | 0.324 | 0.910 |
| RuW2-050 | 9.175 | 7.900 | 0.861 | 0.397 | 0.361 | 0.909 |
| RuW4-050 | 9.560 | 8.100 | 0.847 | 0.419 | 0.373 | 0.890 |

Figure 25 shows the capacity curve of each model. In this figure, the points of displacement limit considering unidirectional (the point at $D_{1U}{}^{*}{}_{uni}$) and bidirectional (the point at $D_{1U}{}^{*}{}_{bi}$) excitations are shown. As is shown in this figure, the difference of the capacity curve of Models RuW1-100 and RuW2-100 (and also Models RuW1-050 and RuW2-050) is significant, while the difference of Models RuW2-100 and RuW4-100 (and also Models RuW2-050 and RuW4-050) is limited.

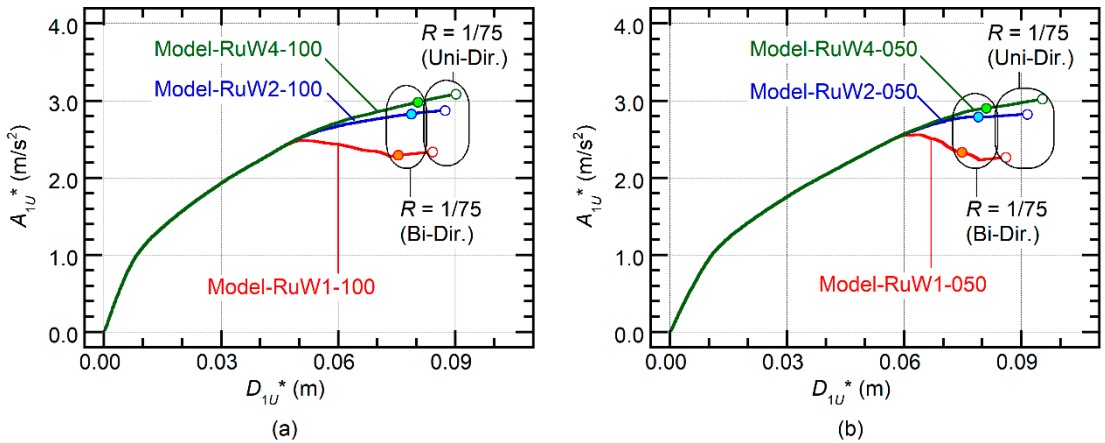

**Figure 25.** Comparison of the capacity curve of each model: (**a**) Models RuW1-100, RuW2-100, and RuW4-100, (**b**) Models RuW1-050, RuW2-050, and RuW4-050.

Figure 26 compares the evaluation results of the capacity index that considers the bidirectional ground motion in each model. As is shown in Figure 26a, the difference between Models RuW1-100 and RuW2-100 is noticeable, while the difference between Models RuW2-100 and RuW4-100 is very small. The same trend can be found in the case of which the stiffness of basement spring is reduced by 50% (Figure 26b).

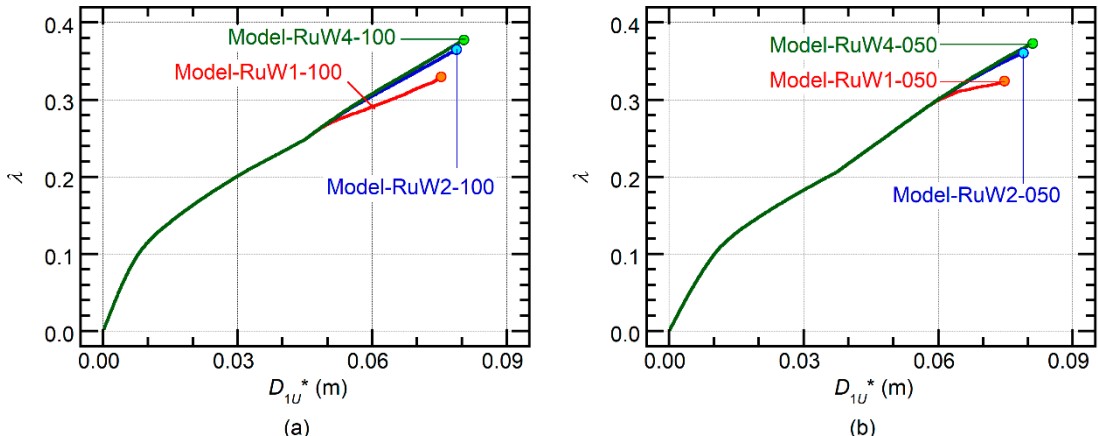

**Figure 26.** Comparison of the evaluated results of the capacity index considering the bidirectional ground motion: (**a**) Models RuW1-100, RuW2-100, and RuW4-100, (**b**) Models RuW1-050, RuW2-050, and RuW4-050.

### 5.2.2. Calculated Peak Story Drift and the Structural Damages

Next, the peak story drift and the structural damages of each model in the case the scaling factor $\lambda = C_{I, bi}$ is calculated according to MABPA. Note that the response spectrum used in the calculation of the peak response of the second modal response is divided by $|\sin\Delta\psi_{12}|$, as is discussed in Section 2.3.4.

Figures 27 and 28 shows the calculated peak story drift of columns $A_1B_1$, $A_3B_1$, and $A_3B_3$. The peak story drift at columns $A_1B_1$ in the second and third story are very close to the limit value ($R = 1/75$), as expected in all models. This is consistent with the preliminary seismic evaluation results shown in Figure 8: the most critical story according to the preliminary evaluation results is the second story. The difference in each model is noticeable in column $A_3B_3$. As shown in Figure 27c, the peak drift of Model RuW1-100 is larger than that of Models RuW2-100 and RuW4-100 in the second story, while in the other stories the peak drift of Model RuW1-100 is smaller than those in other two models. In the case where the stiffness of the basement spring is reduced by 50%, shown in Figure 28c, the peak story drift of Model-RuW1-050 is very close to Models RuW2-050 and RuW4-050 in the second story, while in the other stories the peak drift of Model RuW1-050 is smaller than those in other two models.

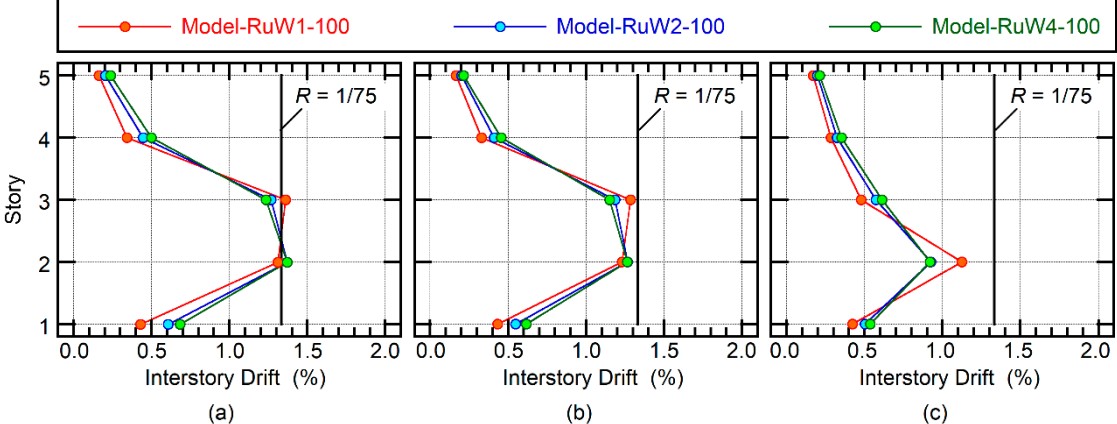

**Figure 27.** Comparison of the peak drift for columns ($\lambda = C_{I, bi}$) for Models RuW1-100, RuW2-100, and RuW4-100: (**a**) column $A_1B_1$, (**b**) column $A_3B_1$, and (**c**) column $A_3B_3$.

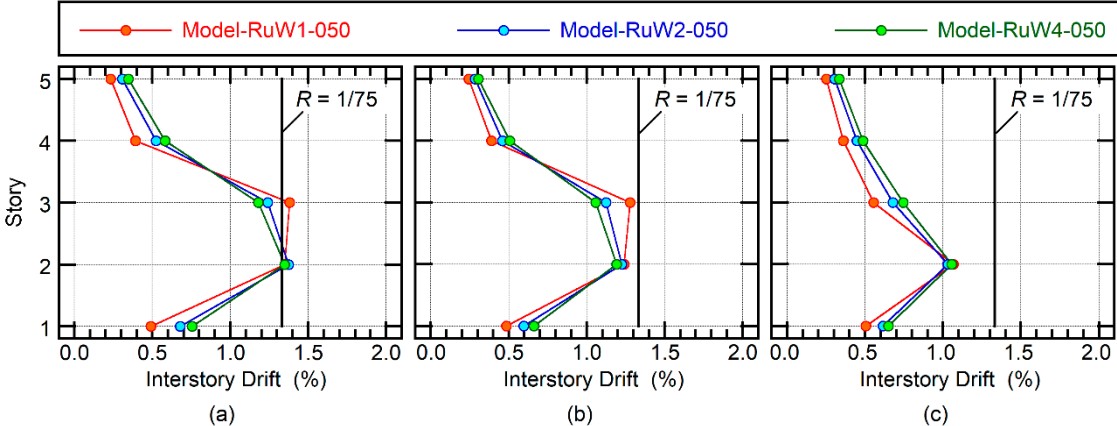

**Figure 28.** Comparison of the peak drift for columns ($\lambda = C_{I,\,bi}$) for Models RuW1-050, RuW2-050, and RuW4-050: (**a**) column $A_1B_1$, (**b**) column $A_3B_1$, and (**c**) column $A_3B_3$.

Figure 29 shows the distribution of the yielding hinges in each model at frames $B_1$ and $A_3$. As shown in this figure, the damage of frame $B_1$ is more significant than $A_3$, in all models: there are no yielding hinges in frame $A_3$ except Models RuW1-100 and RuW4-050, while there are yielding hinges at the end of the beams in levels $Z_2$ and $Z_3$ and the columns in the second and third stories in frame $B_1$. Especially, in case of Models RuW4-100, RuW2-050, and RuW4-050 (Figure 29c,e,f), the flexural yielding occurs at top of column $A_2B_1$ in the third story, and the right-side end of the beam $A_1$-$A_2$ in level $Z_3$ and the left-side end of the beam $A_2$-$A_3$ in level $Z_3$, shown in the dotted rectangle. Note that the yielding strength of the beam ends, where the yielding hinges are made, is reduced because of the yielding of beam-column joint. Therefore, in case of these three models, the yielding of beam-column joint and the column end occur simultaneously at this "+"-shaped beam-column joint.

As shown in Figure 6, the failure of the "+"-shaped beam-column joint at frame $B_1$ was observed above column $A_2B_1$ in the third story, which is consistent to the analysis results of Models RuW4-100, RuW2-050, and RuW4-050. Therefore, it can be concluded the analysis results of the three models may explain the part of the observed damage of the main Uto City Hall building: the reason why the damage in frame $B_1$ was more severe than that in frame $A_3$ is that, due to the effect of torsion, the response of frame $B_1$ is more critical. Considering that these three models may represent a part of the real behavior of this building, the evaluated seismic capacity index of this building is approximately 0.36 to 0.38, as shown in Table 2 (shown in the highlighted).

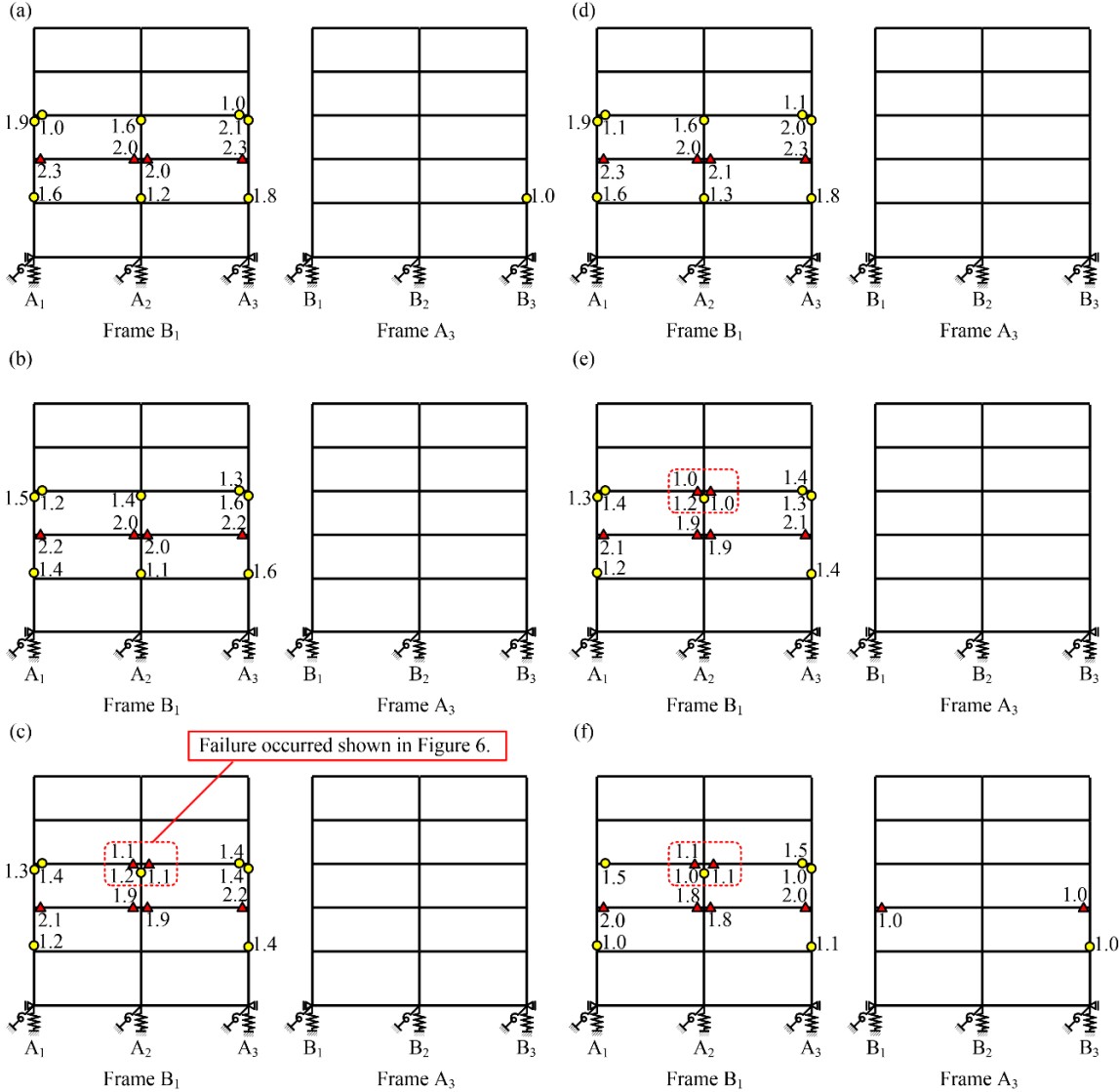

**Figure 29.** Distribution of the yielding hinges at frame $B_1$ and $A_3$ in each model ($\lambda = C_{I,bi}$): (**a**) Model-RuW1-100, (**b**) Model-RuW2-100, (**c**) Model-RuW4-100, (**d**) Model-RuW1-050, (**e**) Model-RuW2-050, and (**f**) Model-RuW4-050.

### 5.2.3. Comparisons with the Evaluated Seismic Capacity and the Response Spectrum of Recorded Earthquakes

Next, the capacity curves of six models are compared to the response spectrum of ground motions recorded during the first (14 April 2016) and second (16 April 2016) earthquakes. The ground motion records used in this analysis were recorded at K-NET Uto station, and details regarding these records can be found in the Appendix C. As described in Section 2, the evaluation of the capacity curve considering bidirectional excitation is based on the assumption that the spectra of the two horizontal components are identical. Of course, this assumption is not true in the real (recorded) ground motion. In this study, the medium of the set of geometrical means obtained using all possible rotations between 0 and 90 degrees (GMRotD50) defined by Boore et al. [44] with the damping ratio equals 0.05 are used as the demand spectrum. The capacity curve of each model is factored by $1/F(_nh_{1eq})$ for the consideration of equivalent damping, as follows:

$$\frac{1}{F\left({}_nh_{1eq}\right)} = \frac{1 + 10{}_nh_{1eq}}{1.5}.$$ 

(34)

Figures 30 and 31 show the comparisons of the factored capacity curve and the demand curve of the first and the second earthquake, respectively.

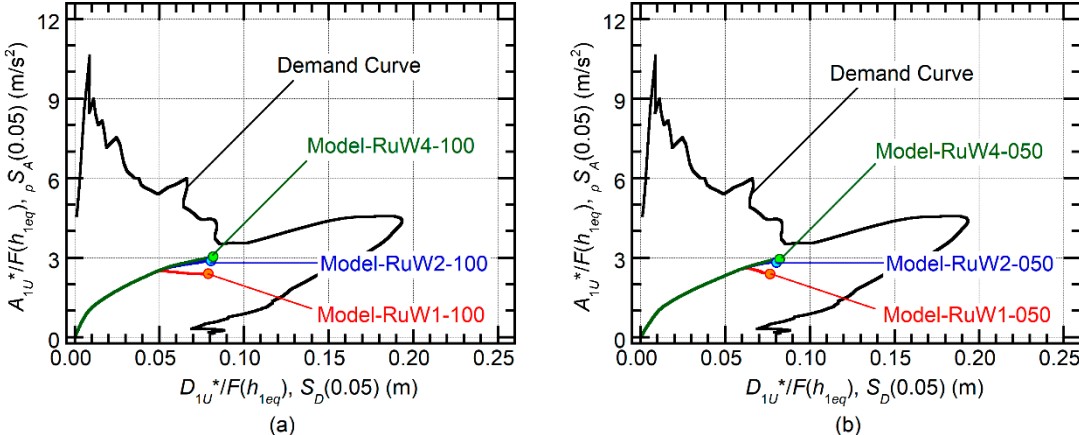

**Figure 30.** Comparison of the factored capacity curve and the demand curve of the first earthquake (14 April 2016): (**a**) Models RuW1-100, RuW2-100, and RuW4-100, and (**b**) Models RuW1-050, RuW2-050, and RuW4-050.

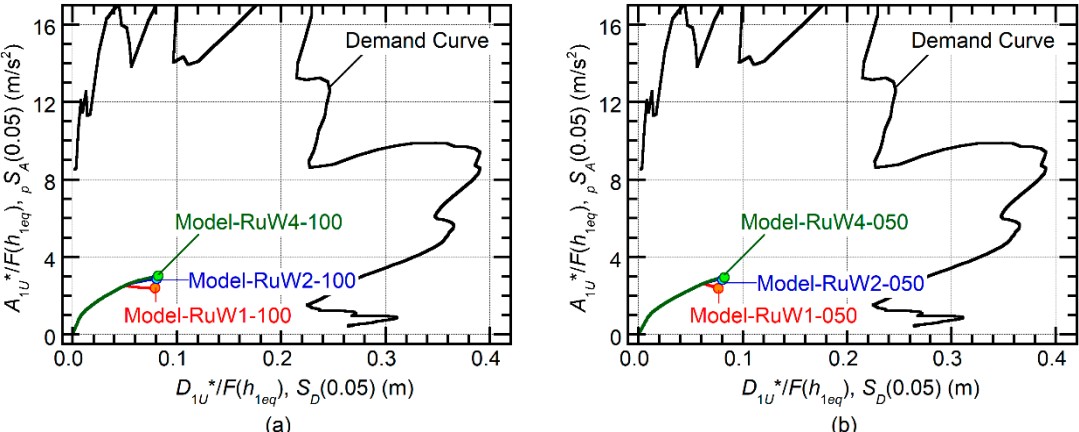

**Figure 31.** Comparison of the factored capacity curve and the demand curve of the second earthquake (16 April 2016): (**a**) Models RuW1-100, RuW2-100, and RuW4-100, and (**b**) Models RuW1-050, RuW2-050, and RuW4-050.

In the case of the first earthquake shown in Figure 30, the capacity curves of all models have no intersection point to the demand curve. Therefore, the response of all building models exceeds the displacement limit $D_{1U}{}^*{}_{bi}$. This implies that the peak drift may exceed 1/75, and the flexural yielding of members shown in Figure 29 may occur during the first earthquake.

Next, in the case of the second earthquake shown in Figure 31, the capacity curves of all models have no intersection point to the demand curve. From the comparisons of two figures, it is seen that the response of all building models during the second earthquake is significantly larger than that during the first earthquake.

*5.3. Discussions*

Since the damages to this building were first reported on the morning of 16 April 2016, after the second earthquake, the most damage to the building had occurred during the second earthquake. However, the results shown in Figure 30 implies that the main Uto City Hall building may suffer some level of damage during the first earthquake, which occurred on 14 April 2016. Based on the results shown in Figures 30 and 31, the author thinks that damage during the first earthquake, especially the simultaneous yielding of the beam-column joint and the top of column $A_2B_1$ in the third story, may be the trigger of the partial collapse in the fourth story occurred during the 16 April earthquake. The simultaneous yielding of the beam-column joint and the top of column $A_2B_1$ in the third story may lead to the unstable behavior of the joint; this may cause the whipping behavior, in which the upper part of building oscillates significantly like the head of a whip, which is when, the author thinks, that the severe damage to the fourth and fifth stories occurred.

## 6. Conclusions

In this paper, the seismic capacity of the main Uto City Hall building, which incurred serious damages during the 2016 Kumamoto Earthquake, was evaluated using the procedure based on the MABPA previously proposed by the author. The main conclusions drawn by this study are as follows.

- A pushover-based procedure to evaluate the seismic capacity index considering bidirectional excitation is presented and its accuracy is verified by the nonlinear time-history analysis. The results show that the accuracy of the evaluated seismic capacity index for this building was satisfactory. Moreover, index $C_{I, bi}$ was better than index $C_{I, uni}$ because in this building, the effect of bidirectional excitation was significant.

- The calculated damages of structural models presented in this paper may explain the part of the observed damage of the main Uto City Hall building during the 2016 Kumamoto Earthquake. The reason why the damage in frame $B_1$ was more severe than that in frame $A_3$ can be explained that, due to the effect of torsion, the response of frame $B_1$ is more critical. Moreover, the severe failure of the beam-column joint at frame $B_1$ observed above column $A_2B_1$ in the third story is consistent with the analysis results of the three analysis models presented in this paper.

- The seismic capacity index of the main Uto City Hall building evaluated by the proposed procedure is approximately 0.36 to 0.38. Therefore, this building could withstand only approximately one-third of the design ground motion specified in the current seismic code of Japan.

- During the first earthquake, which occurred on the 14 April, the main Uto City Hall building may suffer some structural damages, including the yielding of the beam-column joint. According to the comparison of the capacity curve of this building and the demand curve of the first earthquake, the capacity curve has no intersection point to the demand curve.

So far, unfortunately, the ability of the analysis model in this study is only to reveal the possible damage of this building during the first earthquake. Therefore, the author thinks the further improvement will be needed to explain the whole behavior during multiple earthquakes as follows: (i) the modeling of the beam-column joint yielding behavior which can reproduce the unstable behavior of the joint after failure, and (ii) the upgrade of the pushover analysis technique to overcome the limitation of symmetric behavior of all nonlinear springs while keeping numerical stability, applicable for the case when the strength degradation is significant.

It should also be noted that the discussions in Section 5.2.3 rely on the assumption that the GMRotD50 spectrum can represent the bidirectional ground motion in real earthquakes. As the epicenter of the 14 and 16 April earthquakes are not so far from the Uto City Hall (epicentral distance: 15 km and 12 km, respectively [45]), the effect of directivity may not be negligible. Further work will be prepared for the discussion of this issue.

**Author Contributions:** The all contributions according to this article had been made by the first (single) author.

**Funding:** This research received no external funding.

**Acknowledgments:** The author wishes to thank the Uto City Hall officials who provided the original structural designs and other material related to the main buildings. For the analysis in Section 5, the records monitored by National Research Institute for Earth Science and Disaster Prevention were used.

**Conflicts of Interest:** The authors declare no conflict of interest.

## Appendix A. Modification Factor for Predicting the Peak Response of The Equivalent SDOF Model Representing the Second Mode

Figure A1 shows the two equivalent SDOF models representing the first and second modal responses, respectively. Let Equation (A1) be the equation of motion for an *N*-story irregular building model.

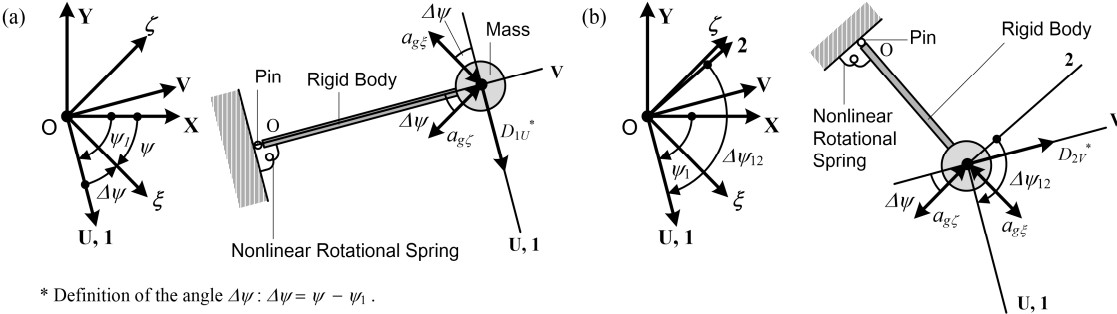

**Figure A1.** Equivalent SDOF model representing each mode: (**a**) first mode and (**b**) second mode.

$$\mathbf{M}\ddot{\mathbf{d}}(t) + \mathbf{C}\dot{\mathbf{d}}(t) + \mathbf{f_R}(t) = -\mathbf{M}\left\{\boldsymbol{\alpha}_\xi a_{g\xi}(t) + \boldsymbol{\alpha}_\varsigma a_{g\varsigma}(t)\right\}, \tag{A1}$$

$$\boldsymbol{\alpha}_\xi = \left\{\begin{array}{ccccccccc} \cos\psi & \cdots & \cos\psi & -\sin\psi & \cdots & -\sin\psi & 0 & \cdots & 0 \end{array}\right\}^{\mathbf{T}}, \tag{A2}$$

$$\boldsymbol{\alpha}_\varsigma = \left\{\begin{array}{ccccccccc} \sin\psi & \cdots & \sin\psi & \cos\psi & \cdots & \cos\psi & 0 & \cdots & 0 \end{array}\right\}^{\mathbf{T}}. \tag{A3}$$

According to previous studies [26,29], the equivalent SDOF model representing the first and second modal response can be expressed as Equations (A4) and (A5), respectively.

$$\ddot{D}_{1U}{}^*(t) + \frac{C_{1U}{}^*}{M_{1U}{}^*}\dot{D}_{1U}{}^*(t) + A_{1U}{}^*(t) = -\left\{\frac{\boldsymbol{\varphi_1}^{\mathbf{T}}\mathbf{M}\boldsymbol{\alpha}_\xi}{\boldsymbol{\varphi_1}^{\mathbf{T}}\mathbf{M}\boldsymbol{\alpha_U}}a_{g\xi}(t) + \frac{\boldsymbol{\varphi_1}^{\mathbf{T}}\mathbf{M}\boldsymbol{\alpha}_\varsigma}{\boldsymbol{\varphi_1}^{\mathbf{T}}\mathbf{M}\boldsymbol{\alpha_U}}a_{g\varsigma}(t)\right\}, \tag{A4}$$

$$\boldsymbol{\alpha}_\varsigma = \left\{\begin{array}{ccccccccc} \sin\psi & \cdots & \sin\psi & \cos\psi & \cdots & \cos\psi & 0 & \cdots & 0 \end{array}\right\}^{\mathbf{T}}. \tag{A5}$$

According to the equivalent SDOF model for the first mode, Equation (A4) can be easily rewritten as follows. Additionally, by considering the relationship of angles $\psi$ and $\psi_1$ (Equation (A6)), Equations (A7) and (A4) can be rewritten as Equation (A8).

$$\frac{\boldsymbol{\varphi_1}^{\mathbf{T}}\mathbf{M}\boldsymbol{\alpha}_\xi}{\boldsymbol{\varphi_1}^{\mathbf{T}}\mathbf{M}\boldsymbol{\alpha_U}} = \cos(\psi - \psi_1) = \cos\Delta\psi, \quad \frac{\boldsymbol{\varphi_1}^{\mathbf{T}}\mathbf{M}\boldsymbol{\alpha}_\varsigma}{\boldsymbol{\varphi_1}^{\mathbf{T}}\mathbf{M}\boldsymbol{\alpha_U}} = \sin\Delta\psi, \tag{A6}$$

$$a_{gU}(t) = a_{g\xi}(t)\cos\Delta\psi + a_{g\varsigma}(t)\sin\Delta\psi, \tag{A7}$$

$$\ddot{D}_{1U}{}^*(t) + \frac{C_{1U}{}^*}{M_{1U}{}^*}\dot{D}_{1U}{}^*(t) + A_{1U}{}^*(t) = -a_{gU}(t). \tag{A8}$$

Equation (A8) is the motion equation representing the first modal response, as has been discussed in previous reports [26,29]. Therefore, no modifications are made according to the first mode. One issue that should be discussed is that the principal axis of the second modal response is not orthogonal

to the U-axis. Let $\Delta\psi_{12}$ be the angle between the principal axes of the first and second modal responses ($\Delta\psi_{12} = \psi_1 - \psi_2$). Then, the following relationship can be obtained:

$$\frac{\boldsymbol{\varphi_2}^{\mathbf{T}}\mathbf{M}\boldsymbol{\alpha_\xi}}{\boldsymbol{\varphi_2}^{\mathbf{T}}\mathbf{M}\boldsymbol{\alpha_V}} = \frac{\cos(\psi - \psi_2)}{\sin(\psi_1 - \psi_2)} = \left(\frac{\cos\Delta\psi_{12}}{\sin\Delta\psi_{12}}\right)\cos\Delta\psi - \sin\Delta\psi, \tag{A9}$$

$$\frac{\boldsymbol{\varphi_2}^{\mathbf{T}}\mathbf{M}\boldsymbol{\alpha_\varsigma}}{\boldsymbol{\varphi_2}^{\mathbf{T}}\mathbf{M}\boldsymbol{\alpha_V}} = \frac{\sin(\psi - \psi_2)}{\sin(\psi_1 - \psi_2)} = \cos\Delta\psi + \left(\frac{\cos\Delta\psi_{12}}{\sin\Delta\psi_{12}}\right)\sin\Delta\psi. \tag{A10}$$

Therefore, by considering Equations (A11) and (A12), Equation (A5) can be rewritten as Equation (A13):

$$-\left\{\frac{\boldsymbol{\varphi_2}^{\mathbf{T}}\mathbf{M}\boldsymbol{\alpha_\xi}}{\boldsymbol{\varphi_2}^{\mathbf{T}}\mathbf{M}\boldsymbol{\alpha_V}}a_{g\xi}(t) + \frac{\boldsymbol{\varphi_2}^{\mathbf{T}}\mathbf{M}\boldsymbol{\alpha_\varsigma}}{\boldsymbol{\varphi_2}^{\mathbf{T}}\mathbf{M}\boldsymbol{\alpha_V}}a_{g\varsigma}(t)\right\} = -\left\{a_{gV}(t) + \frac{\cos\Delta\psi_{12}}{\sin\Delta\psi_{12}}a_{gU}(t)\right\}, \tag{A11}$$

$$a_{gV}(t) = -a_{g\xi}(t)\sin\Delta\psi + a_{g\varsigma}(t)\cos\Delta\psi, \tag{A12}$$

$$\ddot{D}_{2V}{}^*(t) + \frac{C_{2V}{}^*}{M_{2V}{}^*}\dot{D}_{2V}{}^*(t) + A_{2V}{}^*(t) = -\left\{a_{gV}(t) + \frac{\cos\Delta\psi_{12}}{\sin\Delta\psi_{12}}a_{gU}(t)\right\}. \tag{A13}$$

Equation (A13) is identical to the motion equation of the equivalent SDOF model representing the second modal response, which has been proposed by previous studies [26,29], if the principal axis of the second modal response is orthogonal to the U-axis ($\cos\Delta\psi_{12} = 0$). This equation also implies that the contribution of $a_{gU}(t)$ should be considered to estimate the peak response of the second mode, if the principal axis of the second modal response is not orthogonal to the U-axis. In this study, this contribution was considered by factoring the spectral acceleration-displacement curve as follows. Additionally, the factored pseudo acceleration spectrum ($_pS_{AV}{}'(T)$) is defined by Equation (A14).

$$_pS_{AV}{}'(t) = \sqrt{\left\{_pS_{AV}(T)\right\}^2 + \left\{\frac{\cos\Delta\psi_{12}}{\sin\Delta\psi_{12}}{_pS_{AU}(T)}\right\}^2}. \tag{A14}$$

Because the spectra of the two horizontal components were assumed to be identical (as expressed by Equation (1)) Equation (A14) can be rewritten as follows:

$$_pS_{AV}{}'(t) = {_pS_{AU}(T)}\sqrt{1 + \left(\frac{\cos\Delta\psi_{12}}{\sin\Delta\psi_{12}}\right)^2} = \frac{1}{|\sin\Delta\psi_{12}|}{_pS_A(T)}. \tag{A15}$$

Therefore, in this study, the response spectrum used to predict the peak response ($D_{2V}{}^*_{uni}$ and $A_{2V}{}^*_{uni}$) was modified by dividing $|\sin\Delta\psi_{12}|$, as described in Sections 2.3.4 and 2.3.6.

**Appendix B. Building Damage**

Figure A2 shows photographs of the main Uto City Hall building taken after the 16 April earthquake. Figures A3–A5 show the simplified elevations and photographs of each elevation.

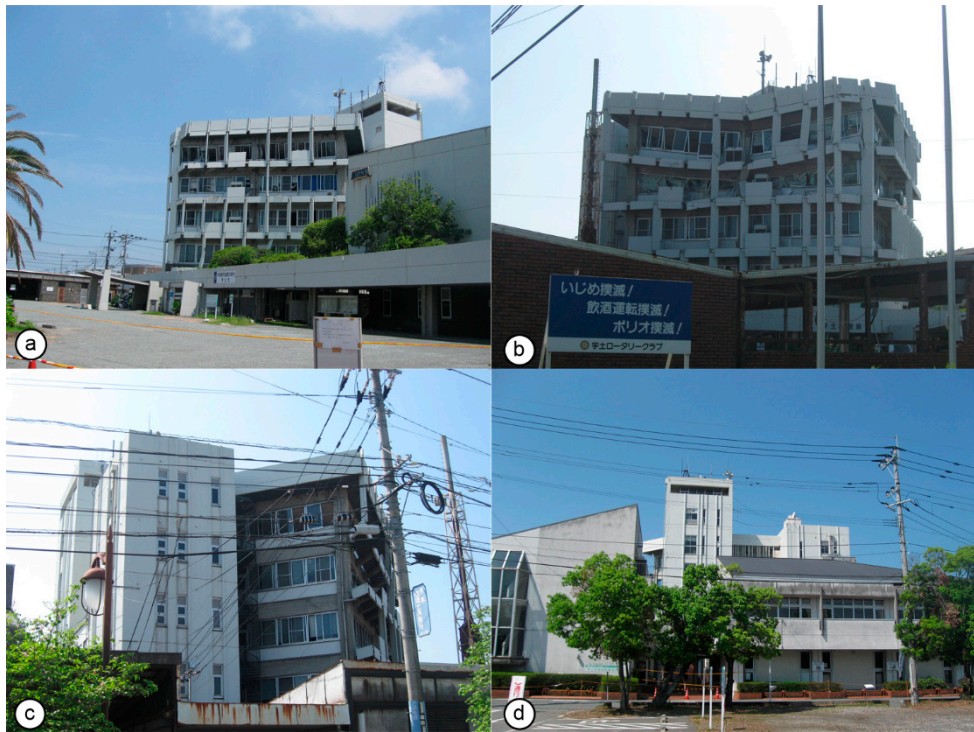

**Figure A2.** Views of the main Uto City Hall building. Photographs were taken from (**a**) the south, (**b**) the west, (**c**) the north, and (**d**) the northeast.

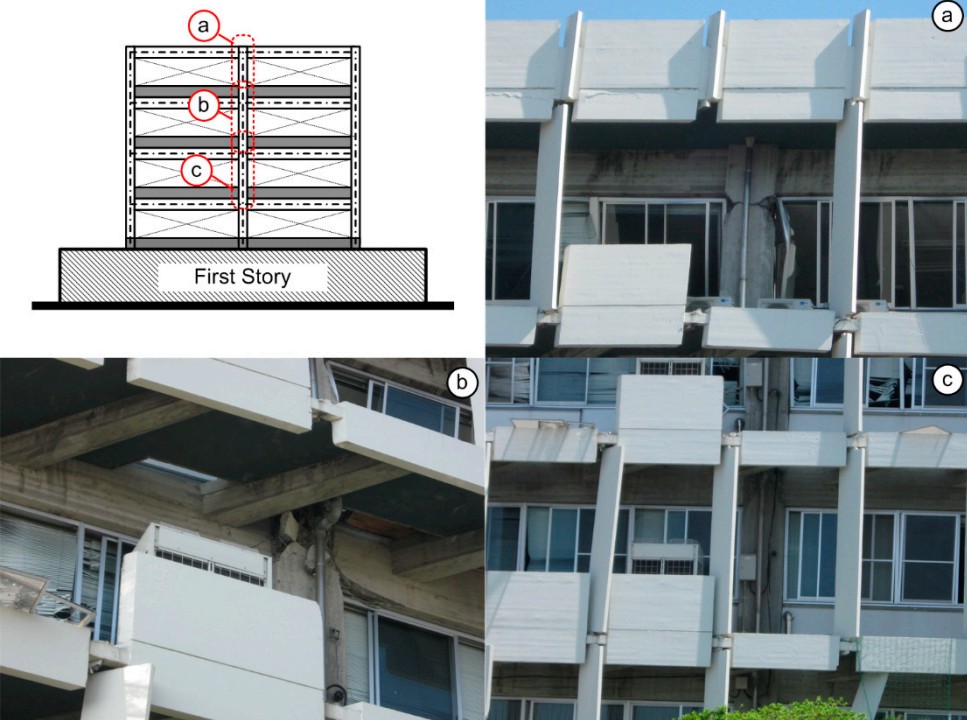

**Figure A3.** Simplified elevation and photographs of the south elevation. Photographs show damage to (**a**) a fifth-story column, (**b**) a beam–column joint at the top of the fourth story, and (**c**) a third story column.

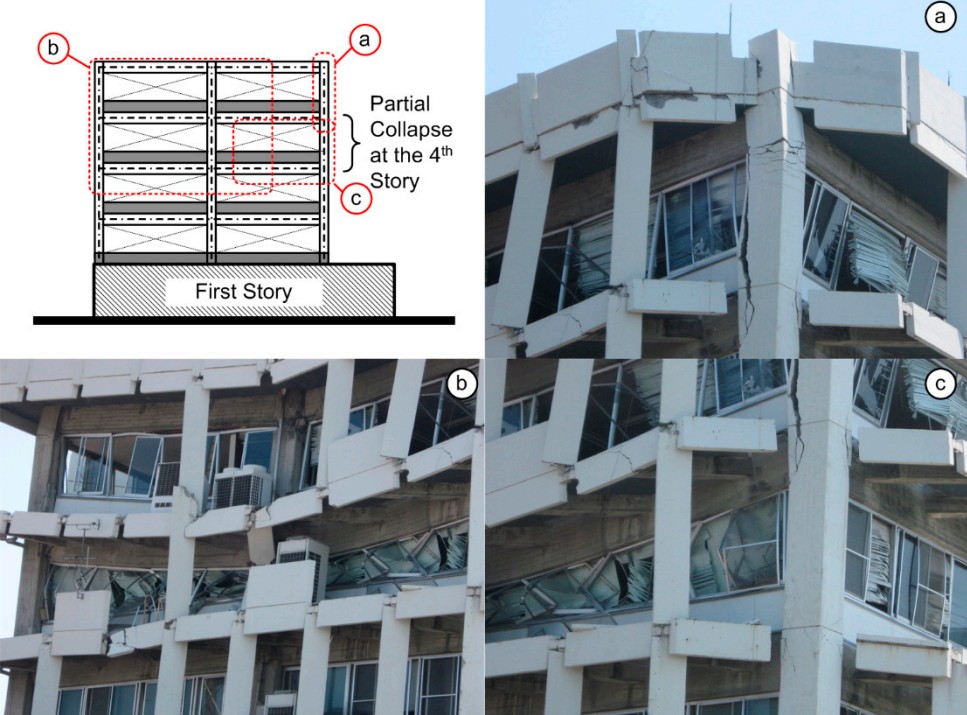

**Figure A4.** Simplified elevation and photographs of the west elevation. Photographs show (**a**) damage to the corner column on the fifth story, (**b**) partial collapse of the fourth story, (**c**) damage to the corner column on the fourth story.

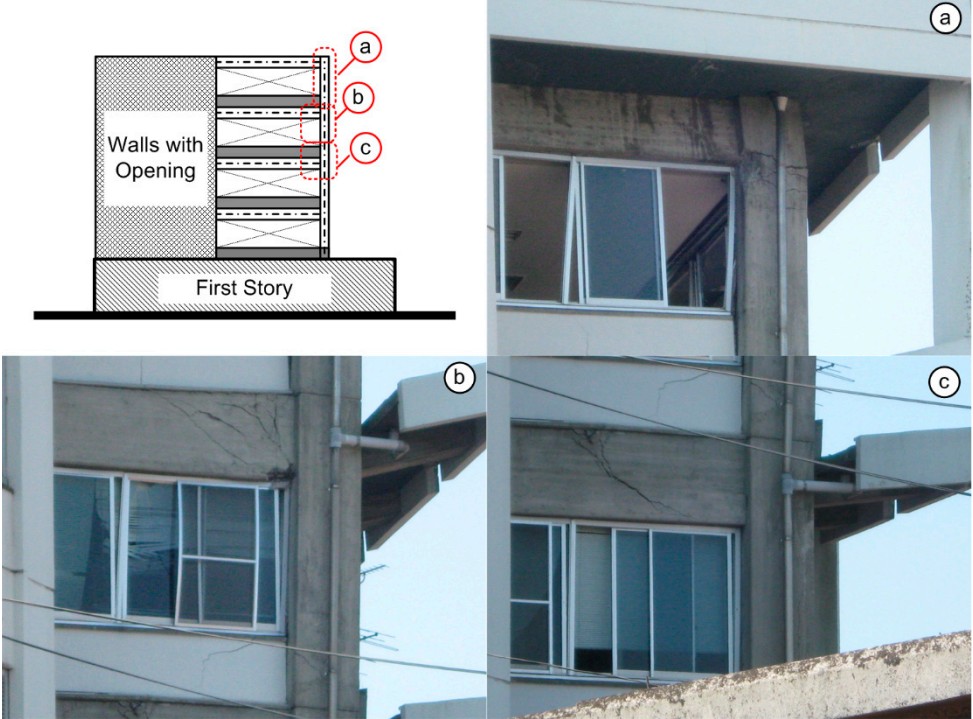

**Figure A5.** Simplified elevation and photographs of the north elevation. Photographs show damage to (**a**) the top of corner columns and beams on the roof floor, (**b**) a beam on the fifth floor, and (**c**) the bottom of a fourth-story column and a fourth-floor beam.

### Appendix C. Record Observed in the 14th and 16th Earthquake at K-NET Uto Station

Figures A6 and A7 show the acceleration of two horizontal components and orbit of the record observed on the 14 and 16 April, respectively, of the April earthquake at the K-NET Uto Station. The major and minor axes in the orbit are determined according to Penzien and Watabe [46]. Figure A8 shows the pseudo-acceleration response spectrum of the two recorded ground motions observed at the K-NET Uto Station.

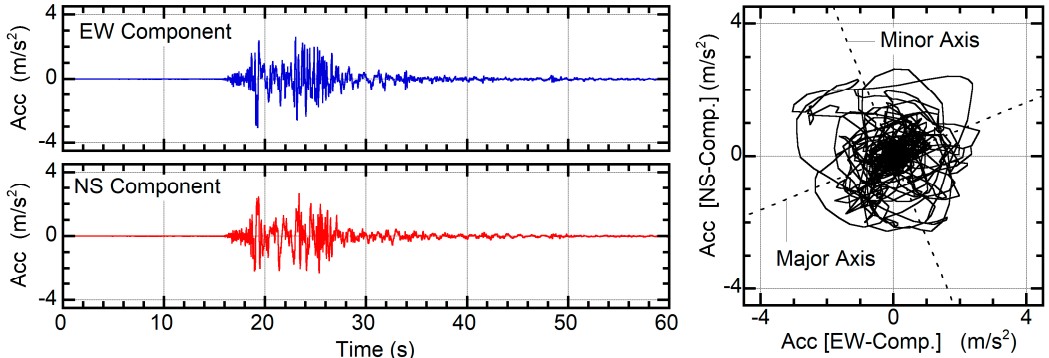

**Figure A6.** Acceleration of EW- and NS-components and orbit of the record observed in the 14 April earthquake at the K-NET Uto Station.

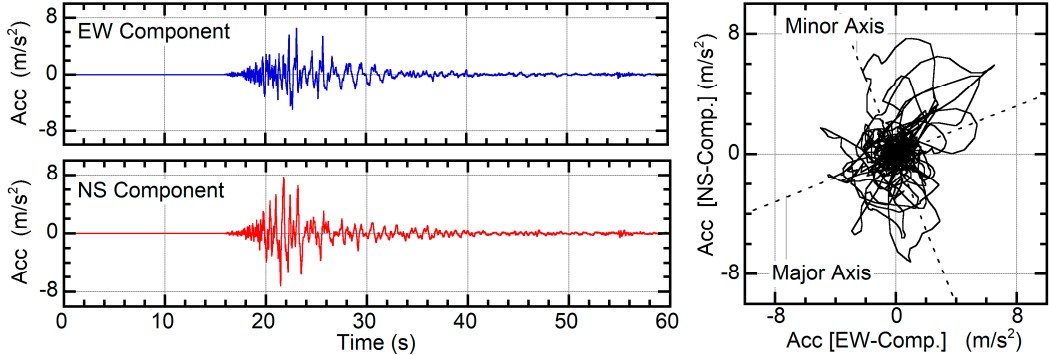

**Figure A7.** Acceleration of EW- and NS-components and orbit of record observed in the 16 April earthquake at the K-NET Uto Station.

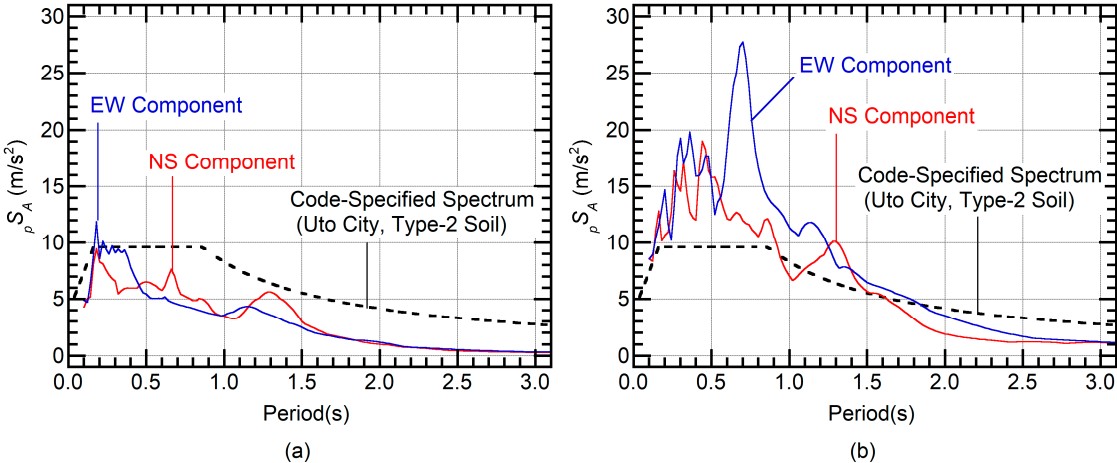

**Figure A8.** Pseudo acceleration response spectrum of two recorded ground motion at the K-NET Uto Station: (**a**) 14 April earthquake, and (**b**) 16 April earthquake.

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
