# Peer review of "Pushover-Based Seismic Capacity Evaluation of Uto City Hall Damaged by the 2016 Kumamoto Earthquake"

_buildings, doi:10.3390/buildings9060140_

Round 1
Reviewer 1 Report
Pushover-Based Seismic Capacity Evaluation of Uto City Hall Damaged by the 2016 Kumamoto Earthquake
In this paper the seismic capacity of Uto City Hall building which was damaged by Kumamoto Earthquake is presented using the pushover method. The comments below will clarify the authors’ results and give to the readers a better understanding of the material presented.
Review comments
1. The objectives of this paper need to be further addressed.
2. The authors should give guidelines of how an engineer should proceed to evaluate the seismic capacity of a structure using the method presented.
3. A better description of the building should be given, together with the main damages that occurred (without having to refer to another paper).
4. The authors should mention the criteria that must be used to select the blocks that needed to model a structure of different shape for analysis purposes.
5. The authors should explain what are the advantages of using the pushover method instead for example the finite element method which can give a better overview of the weak parts of the structure easily and can be analyzed for different excitations?
6. Correct the English mistake on lines 754-755 ‘…can be explained the response…’
Author Response
Please check PDF file.

Reviewer 2 Report
Congratulations for your work! It is very well designed and comprehensively explained.
Nevertheless I would suggest rewriting (restructuring) all the introduction, and especially section 1.2. Brief review of related studies as it is – the most of it – taken from another paper that you have already accomplished ( Fujii (2018) - Prediction of the peak seismic response of asymmetric buildings under bidirectional horizontal ground motion using equivalent SDOF model). Somebody could see it as a self-plagiarism.
The discussions could also be more detailed, also including other points of view/hypoteses...
For the rest, I believe, the paper is high quality research. Nevertheless a second check from another reviewer more specialized in the technical aspects of (physical) vulnerability would be necessary.
Author Response
Please check the uploaded PDF file about the comment.

Reviewer 3 Report
The manuscript presents a study on the seismic capacity assessment of the main Uto City Hall building, which was damaged by the 2016 Kumamoto Earthquake in Japan. The review of pushover procedures, the subsequent application of the pushover-based method (including the effect of bidirectional excitation) developed by the Author in a previous work, and the validation of the assessment procedure by nonlinear time-history analysis, are an overall valuable contribution with relevance for practice. The background and objectives of the work are well described and the research methodology is suitable and supported by the relevant references. There are however some concepts which should be further explained in the text, e.g. meaning of the ‘scaling factor’. The manuscript is in general well structured and written. The results are clearly presented and the relevant conclusions are provided. Some minor improvements are proposed below:
1. Section 2.2.1: A figure can help to better comprehend the presented formulation, e.g. to elucidate about the involved angles.
2. Figure 7 presents a lot of curves/formulas which are difficult to realize. Please provide a better description in the text.
3. Line 436: The correct mention should be ‘The envelope shown in Figure 10 (b)’.
4. Line 581: The Authors should provide the accelerograms of the artificial ground motions.
5. Line 633: There is a typo with the ‘Thus’. Figure 27: What the values presented next to the hinges are?
6. Line 740: The term ‘whipping behavior’ is not so clear. It may refer to the ‘chain effect’? Please make clear.
7. The headings of Appendices A and B should be consistent. Figures A2 to A4 should be renumbered to B1 to B3.
8. Reference 9 to ASCE/SEI 41-06 should be updated to the last version of the equivalent document, i.e. ASCE/SEI 41-17.
Author Response
Please check the PDF file.
